# IL-2/JES6-1 mAb complexes dramatically increase sensitivity to LPS through IFN-γ production by CD25+Foxp3- T cells

Jakub Tomala[1], Petra Weberova[1], Barbora Tomalova[1],
Zuzana Jiraskova Zakostelska[2], Ladislav Sivak[1], Jirina Kovarova[1], Marek Kovar[1]*

[1]Laboratory of Tumor Immunology, Institute of Microbiology, Czech Academy of Sciences, Prague, Czech Republic; [2]Laboratory of Cellular and Molecular Immunology, Institute of Microbiology, Czech Academy of Sciences, Prague, Czech Republic

**Abstract** Complexes of IL-2 and JES6-1 mAb (IL-2/JES6) provide strong sustained IL-2 signal selective for CD25+ cells and thus they potently expand $T_{reg}$ cells. IL-2/JES6 are effective in the treatment of autoimmune diseases and in protecting against rejection of pancreatic islet allografts. However, we found that IL-2/JES6 also dramatically increase sensitivity to LPS-mediated shock in C57BL/6 mice. We demonstrate here that this phenomenon is dependent on endogenous IFN-γ and T cells, as it is not manifested in IFN-γ deficient and nude mice, respectively. Administration of IL-2/JES6 leads to the emergence of CD25+Foxp3-CD4+ and CD25+Foxp3-CD8+ T cells producing IFN-γ in various organs, particularly in the liver. IL-2/JES6 also increase counts of CD11b+CD14+ cells in the blood and the spleen with higher sensitivity to LPS in terms of TNF-α production and induce expression of CD25 in these cells. These findings indicate safety issue for potential use of IL-2/JES6 or similar IL-2-like immunotherapeutics.

## Editor's evaluation

Tomala et al., describes the ability of IL-2/JES6-1 mAb complexes to increase mouse sensitivity to LPS challenge. The authors present data to suggest this is due to IFN-γ production by CD25+Foxp3- T cells. The manuscript has identified an interesting phenomenon as a result of IL-2/JES6-1 complex administration. These data may provide novel avenues for future therapeutic intervention in autoimmune disease.

*For correspondence:
makovar@biomed.cas.cz

**Competing interest:** The authors declare that no competing interests exist.

## Introduction

IL-2 is a crucial cytokine for activation, expansion, and expression of effector functions of T and NK cells as well as for homeostasis of regulatory T ($T_{reg}$) cells (*Boyman and Sprent, 2012*; *Liao et al., 2013*; *Malek and Castro, 2010*). IL-2 exerts its biological activities through binding to either a dimeric receptor composed of IL-2Rβ (CD122) and common cytokine receptor gamma chain ($γ_c$, CD132) or to a trimeric receptor composed of a dimeric one with the addition of IL-2Rα (CD25) (*Minami et al., 1993*; *Waldmann, 1989*). CD25 is not involved in signal transduction, but increases the affinity of the trimeric receptor for IL-2 by about 100-fold (*Minami et al., 1993*; *Waldmann, 1989*). The dimeric receptor ($K_d$ ~1 nM) is mostly found on memory CD8+ T and NK cells, whilst the trimeric receptor ($K_d$ ~10 pM) is typically expressed at high levels by $T_{reg}$ cells, recently activated T cells and ILC2s (*Boyman and Sprent, 2012*; *Roediger et al., 2013*; *Sakaguchi et al., 1995*).

IL-2 was the first immunotherapy approved for the treatment of metastatic renal cell carcinoma and malignant melanoma in the early 1990s (*Atkins et al., 1999*; *Klapper et al., 2008*), having induced

an overall response in 15–17% of patients with these cancers including 5–10% durable complete remissions (*Atkins et al., 1999*; *Klapper et al., 2008*). However, an extremely short half-life (*Donohue and Rosenberg, 1983*) and serious side toxicities associated with high-dose IL-2 treatment are the major drawbacks. Moreover, IL-2 treatment leads to expansion of T$_{reg}$ cells which can dampen effector T cell activity against tumors (*Berendt and North, 1980*). Indeed, it has been shown that long-term, low-dose IL-2 therapy preferentially stimulates T$_{reg}$ cells due to their constitutive high expression of trimeric IL-2R and can be used for treatment of autoimmune diseases and delaying allograft rejection (*Klatzmann and Abbas, 2015*; *Yu et al., 2009*). However, due to significant limitations of IL-2, strong attempts have been made to improve IL-2-based therapy. One of the most promising approaches is to employ complexes of IL-2 and certain anti-IL-2 mAbs. It has been demonstrated that these complexes not only markedly prolong the half-life of parenterally administered IL-2 (from minutes to hours) but they also exert selective stimulatory activity for different immune cell populations depending on the mAb used (*Boyman et al., 2006*; *Tomala et al., 2009*). Complexes of murine IL-2 and S4B6 mAb (IL-2/S4B6 henceforth) were shown to potently stimulate memory CD8$^+$ T and NK cells, whilst complexes of murine IL-2 and JES6-1 mAb (IL-2/JES6 henceforth) highly selectively expand T$_{reg}$ cells (*Boyman et al., 2006*; *Spangler et al., 2015*; *Tomala et al., 2009*). Selectivity of IL-2/JES6 for cells expressing the high-affinity IL-2R is governed by the fact that the binding site for JES6-1 mAb and CD122/CD132 in the IL-2 molecule, almost completely overlap. IL-2/JES6 is thus not able to bind to IL-2R and induce signaling per se. However, CD25 can engage IL-2 bound to JES6-1 mAb, particularly when expressed at high levels, and progressively 'peel off' the cytokine from the antibody in a zipper-like mechanism, ultimately leading to dissociation of JES6-1 mAb. Once JES6-1 mAb dissociates, the CD25-bound IL-2 is liberated to recruit CD122/CD132 to form the functional signaling complex (*Spangler et al., 2015*). Selectivity of IL-2/JES6 for CD25$^+$ cells results in their high efficacy to expand T$_{reg}$ cells in vivo and makes them an attractive immunotherapeutic. It has been shown that IL-2/JES6 could be used for treatment of various autoimmune diseases (*Izquierdo et al., 2018*; *Liu et al., 2011*; *Webster et al., 2009*; *Wilson et al., 2008*) and to facilitate long-term acceptance of allografts without the need for immunosuppression (*Webster et al., 2009*). A single-chain format of IL-2/JES6 was produced with a mutated JES6-1 mAb-binding site, termed JY3, which affected the affinity for IL-2. This proved to be effective in selective expansion of T$_{reg}$ cells and in the model of autoimmune colitis (*Spangler et al., 2018*). Furthermore, there is a report describing the development of fully human mAb binding human IL-2 with the capacity to selectively expand T$_{reg}$ cells in vivo when complexed with human IL-2 (*Trotta et al., 2018*).

It has been shown that T$_{reg}$ cells expanded by IL-2/JES6 produce IL-10 and TGF-β (*Webster et al., 2009*). We therefore decided to explore the possibility that IL-2/JES6 are able to protect against LPS-induced toxicity. Interestingly, we found that short-term pre-treatment (three daily doses) with IL-2/JES6 dramatically increases sensitivity of C56BL/6 mice to LPS-induced shock and mortality. Thus, we decided to further investigate this phenomenon and to uncover the mechanism responsible for IL-2 signal-mediated LPS hyperreactivity.

## Results
### IL-2/JES6, but Not IL-2/S4B6, dramatically increase sensitivity to LPS

We asked whether selective expansion of T$_{reg}$ cells via treatment with IL-2/JES6 could lead to protection from the toxic effect of LPS. To answer this we injected C56BL/6 mice with three daily doses of IL-2/JES6 (1.5 µg IL-2/dose) and challenged these mice with LPS 48 h after the last dose of IL-2/JES6 (*Figure 1A*). We titrated LPS dosing in previous experiments in order to determine the maximum non-lethal dose (MNLD), that is, the highest dose that causes significant toxic effect but no mortality. Since we found that toxicity of LPS varies from batch to batch even if ordered as identical product from the same commercial supplier, we isolated a large batch of LPS from *S. typhymurium* LT2, S-strain (see Materials and methods) and used it throughout the whole study. We determined that the MNLD of our LPS was ~200 µg/mice in C56BL/6 mice. Control C56BL/6 mice developed hypothermia starting about 4–6 h after LPS challenge (100% of MNLD) and peaking after approximately 24 h. However, all mice recovered over the next 2 d. Mice injected with a low dose of LPS (10% of MNLD) developed only negligible hypothermia 8 h after LPS injection and fully recovered within 24 h. Contrary to that, the same low dose of LPS induced extremely rapid onset of progressively worsening hypothermia

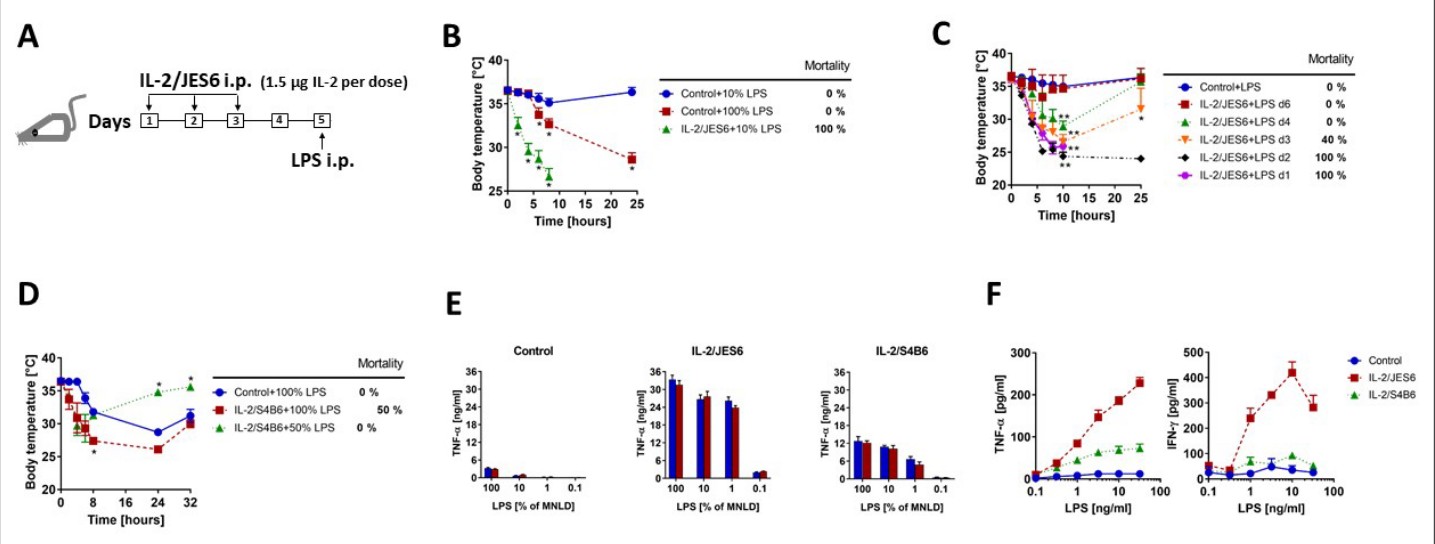

**Figure 1.** IL-2/JES6 dramatically increase sensitivity to LPS-induced shock and mortality. (**A**) Schedule of sensitization of mice to LPS through administration of IL-2/JES6 used throughout the study, unless stated otherwise. (**B**) C56BL/6 mice were treated with IL-2/JES6 as shown in A. Control mice were treated with sterile PBS. The dose of injected LPS is shown in % of maximum non-lethal dose, henceforth (MNLD; 100% ~ 200 µg LPS/mice). (**C**) C56BL/6 mice were treated with IL-2/JES6 and LPS (10% of MNLD) was injected 1–6 d after the last dose of IL-2/JES6. (**D**) C56BL/6 mice were treated with complexes of IL-2 and S4B6 mAb (IL-2/S4B6) followed by LPS challenge (100 or 50% of MNLD) using the same schedule as in A. (**E**) C56BL/6 mice were treated as in A with PBS (Control), IL-2/JES6 or IL-2/S4B6 complexes. Mice were euthanized 90 min post administration of titrated doses of LPS and their individual sera were collected. Concentration of TNF-α in the serum was determined by ELISA. Each bar represents one individual mouse± SD (n = 3 technical replicates). (**F**) C56BL/6 mice were treated as in A with PBS (Control), IL-2/JES6 or IL-2/S4B6 complexes, but not challenged with LPS. Mice were euthanized 2 d after the last dose of complexes and their spleen cells were cultivated in titrated concentrations of LPS for 16 h in vitro. Concentrations of TNF-α and IFN-γ in the supernatant were determined by ELISA. Each point represents pool of three individual mouse ± SD (n = 3 technical replicates). All experiments were done at least twice with similar results; n = 4–7 technical replicates (**B–D**). Data were analysed using an unpaired two-tailed Student's t-test. Significant differences to control are shown (* p ≤ 0.05; ** p ≤ 0.01).

The online version of this article includes the following source data and figure supplement(s) for figure 1:

**Source data 1.** Source data for *Figure 1*, panels B-F.

**Figure supplement 1.** Nitrite significantly reduces toxicity of LPS in mice pretreated with IL-2/JES6.

**Figure supplement 1—source data 1.** Source data for *Figure 1—figure supplement 1*.

**Figure supplement 2.** IL-2/JES6 induce more severe lung oedema in comparison to IL-2/S4B6.

**Figure supplement 2—source data 1.** Source data for *Figure 1—figure supplement 2*.

when mice were pretreated with IL-2/JES6 and the mice usually died within 8–24 h (*Figure 1B*). Next, we decided to determine the kinetics of sensitization to LPS by IL-2/JES6. Mice pretreated with IL-2/JES6 as in *Figure 1A* were challenged at different time points after IL-2/JES6 treatment with low dose of LPS (10% of MNLD). *Figure 1C* shows that LPS caused 100% mortality when injected up to 2 d post IL-2/JES6 treatment and 40% mortality when injected 3 d after that. Significant hypothermia, but no mortality was seen when LPS was injected 4 d post IL-2/JES6. No sensitization to LPS was found when LPS was injected 6 d post IL-2/JES6. These data shows that IL-2/JES6 dramatically increase sensitivity to LPS in C56BL/6 mice and that this increased sensitivity lasts about 4 d post IL-2/JES6 treatment.

We also wanted to know whether IL-2/S4B6 sensitize C56BL/6 mice to LPS. Mice pretreated with IL-2/S4B6 using the same schedule as in *Figure 1A*, showed only decent sensitization to LPS as challenge with relatively high dose of LPS (50% of MNLD) induced hypothermia but no mortality (*Figure 1D*). Thus, IL-2 complexes with selective stimulatory activity for CD25[high] cells, but not those which stimulate predominantly CD122[high] cell populations, dramatically increase sensitivity to LPS. Accordingly, TNF-α levels in sera of mice pretreated with IL-2/JES6 were significantly higher in comparison to sera from mice pretreated with IL-2/S4B6 after LPS challenge (*Figure 1E*). Furthermore, splenocytes from IL-2/JES6 pretreated mice produced much higher amounts of TNF-α and IFN-γ than those from IL-2/S4B6 pretreated mice when cultivated in the presence of LPS in vitro (*Figure 1F*).

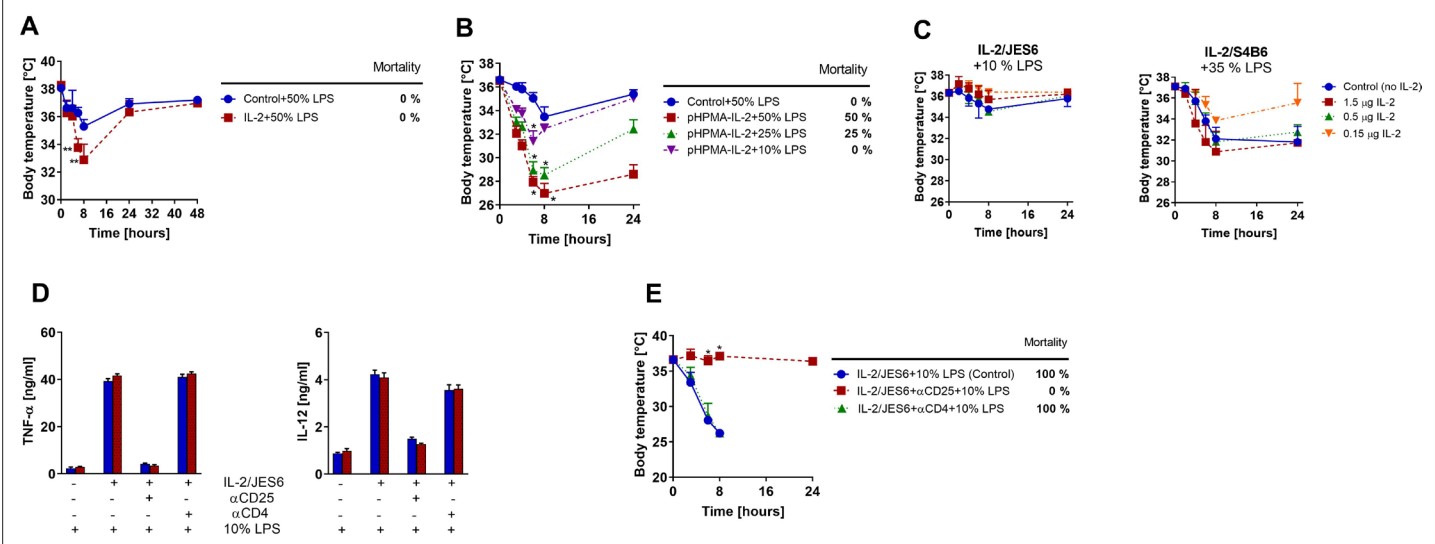

**Figure 2.** Strong sustained IL-2 signal and not solely IL-2/JES6, increases sensitivity to LPS; IL-2/JES6-mediated sensitivity to LPS could be completely blocked by αCD25 mAb. (**A**) C56BL/6 mice were treated with rmIL-2 (35 µg/dose) according to the schedule shown in *Figure 1A* and challenged with LPS (50% of MNLD). (**B**) C56BL/6 mice were treated with IL-2 covalently bound to the polymeric carrier based on poly(HPMA) using the same schedule as in *Figure 1A* and subsequently challenged with titrated doses of LPS. (**C**) C56BL/6 mice were injected simultaneously with LPS and either IL-2/JES6 or IL-2/S4B6 complexes (1.5 or 0.5 or 0.15 µg IL-2/dose) in one i.p. injection. Dosage of LPS is shown in % of MNLD above each graph. Control mice were injected with the same dose of LPS only. (**D**) C56BL/6 mice were injected with IL-2/JES6 and then challenged with LPS (10% of MNLD) as shown in *Figure 1A*. Some mice were also injected with either αCD25 or αCD4 mAb (250 µg/mouse i.p.) 4 h prior to the first injection of IL-2/JES6 as shown below in the graphs. Mice were euthanized 4 h after the LPS challenge and concentrations of TNF-α and IL-12 were determined by ELISA. Each bar represents one individual mouse± SD (n = 3 technical replicates). (**E**) C56BL/6 mice were treated with IL-2/JES6 and subsequently challenged with LPS as shown in *Figure 1A*. Some mice were also injected with either αCD25 or αCD4 mAb (250 µg per mouse i.p.) 4 h prior to the first injection of IL-2/JES6. All experiments were done at least twice with similar results; n = 4–6 technical replicates. Data were analysed using an unpaired two-tailed Student's t-test. Significant differences to control are shown (* p ≤ 0.05; ** p ≤ 0.01).

The online version of this article includes the following source data for figure 2:

**Source data 1.** Source data for *Figure 2*, panels A-E.

## Both durability of IL-2 signal and its selectivity for CD25[high] cells are necessary to induce high sensitivity to LPS

Next, we asked whether IL-2 alone is also able to measurably increase the sensitivity of C56BL/6 mice to LPS. We pretreated mice with high IL-2 dosage (35 µg/dose, i.e., more than 20-times higher compared to IL-2/JES6) and challenged them with LPS using the same schedule as in *Figure 1A*. We found that mice pretreated with IL-2 developed more profound hypothermia than controls, however, no mortality was recorded (*Figure 2A*). Thus, IL-2 alone can produce a slight increase in sensitivity to LPS when a high dosage is used. We decided to test whether IL-2 bound to a biocompatible polymeric carrier based on poly(*N*-(2-propyl)methacrylamide) would be more effective in sensitization to LPS. The rationale here is that IL-2 bound to this polymeric carrier has a significantly prolonged half-life in circulation (3–4 h, i.e., similar to IL-2 complexes) thus providing a more sustained IL-2 signal but with no selectivity for CD25[high] cells (*Votavova et al., 2015*). Indeed, IL-2 bound to a polymeric carrier sensitized C56BL/6 mice to LPS substantially more effectively than IL-2. The LPS challenge with 50% of MNLD caused more severe hypothermia and 50% mortality (*Figure 2B*). These data demonstrate that durability of the IL-2 signal plays a remarkable role in sensitization to LPS via IL-2. However, sensitivity to LPS induced by IL-2 bound to the polymeric carrier is still much weaker in comparison to IL-2/JES6, showing that selectivity for CD25[high] cells is also important. We injected LPS 48 h after the last dose of IL-2 complexes in our sensitization experiments. Therefore, it is highly unlikely that there was still a biologically active concentration of these complexes at the time of LPS administration since the half-life of IL-2 complexes was determined to be several hours. However, to exclude the possibility that increased sensitivity to LPS is caused by the presence of IL-2 complexes at the time of LPS injection, that is, that LPS and IL-2 complexes acts together at the same time, we injected C56BL/6 mice with

either IL-2/JES6 or IL-2/S4B6 and simultaneously challenged them with LPS (10% or 35% of MNLD, respectively). Simultaneous co-administration of IL-2 complexes and LPS did not increase toxicity of LPS, as seen in *Figure 2C*. These data show that it is important the IL-2/JES6 to be present before LPS administration rather than at the time of administration in order to induce LPS hyperreactivity. Thus, IL-2/JES6 either act on cells that are able to directly respond to LPS making them hyperresponsive, or they activate some immune cell subset(s), which in turn mediates LPS hyperreactivity, for example, via production of some effector molecule(s).

Since high expression of CD25 is a prerequisite for the ability to utilize IL-2/JES6, we decided to test whether blocking of CD25 abrogates sensitization to LPS by IL-2/JES6. Indeed, administration of anti-CD25 mAb completely diminished sensitization to LPS by IL-2/JES6 measured by both TNF-α or IL-12 serum levels (*Figure 2D*) and hypothermia plus mortality (*Figure 2E*). Notably, administration of anti-CD4 mAb had no effect, although CD25$^{high}$ cells are mostly T$_{reg}$ cells in naïve unprimed mice (*Figure 2D and E*). This show that CD4$^+$ T cells, including T$_{reg}$ cells, do not play an irreplaceable role in the studied phenomenon.

## IL-2/JES6 increase counts of CD11b$^+$CD14$^+$ cells and their responsiveness to LPS

Our next set of experiments aimed to deduce how IL-2/JES6 affect LPS-responding myeloid cells, particularly CD11b$^+$CD14$^+$ cells. We found that treatment of C56BL/6 mice with IL-2/JES6 (as in *Figure 1A*) increased relative counts of CD11b$^+$CD14$^+$ cells in the spleen and blood (*Figure 3A and B*) as well as absolute counts of these cells in the spleen (*Figure 3C*). We also found that treatment with IL-2/JES6 expands myeloid cells in general. IL-2/JES6 expanded significantly granulocytes, eosinophils and DCs and elevated, though not significantly, relative counts of monocytes and macrophages (*Figure 3—figure supplement 1*). Of note, treatment with IL-2/JES6 increased MHC II expression on monocytes and macrophages (*Figure 3—figure supplement 2*). The proliferation of myeloid cells driven by IL-2/JES6 was further confirmed by BrdU incorporation (*Figure 3—figure supplement 3*). IL-2/JES6 also remarkably increased responsiveness of CD11b$^+$CD14$^+$ cells from the spleen (*Figure 3D and E*) and blood (*Figure 3F and G*) to LPS in term of TNF-α production. Moreover, we found that IL-2/JES6 increased expression of CD25 in CD11b$^+$CD14$^+$ cells from the spleen and liver (*Figure 3H, I*) thus enabling these cells to effectively utilize IL-2/JES6. These data collectively show that treatment with IL-2/JES6 increases both the counts of LPS-responsive myeloid cells in various body compartments and their responsiveness to LPS. We asked a question whether this increased responsiveness of myeloid cells to LPS was due to the increased expression of TLR4. Thus, we analyzed *Tlr4* expression in spleen cells via quantitative RT-PCR and in various myeloid cell subsets via flow cytometry. Treatment with IL-2/JES6 did not affect the *Tlr4* expression in splenocytes on mRNA level (*Figure 3—figure supplement 4*). No statistically significant difference in TLR4 level upon IL-2/JES6 treatment in comparison to control was found in CD11b$^+$Ly6G$^-$Ly6C$^{high}$ cells. Surprisingly, IL-2/JES6 treatment decently but statistically significantly decreased TLR4 levels in CD11b$^+$Ly6G$^+$ cells and CD11b$^+$Ly6G$^-$Ly6C$^{low}$cells (*Figure 3—figure supplement 5*). IL-2/JES6 thus did not increased the sensitivity to LPS via increased expression of TLR4.

## T cells and IFN-γ are essential for sensitization to LPS by IL-2/JES6

IFN-γ is known to activate myeloid cells, particularly monocytes and macrophages, to more vigorously respond to TLR ligands including LPS. Thus, we decided to compare sensitization to LPS by IL-2/JES in C56BL/6 mice and BALB/c mice. C56BL/6 mice are described as a strand with sufficient IFN-γ production while BALB/c mice are poor producers. We determined levels of various cytokines in sera of C56BL/6 and BALB/c mice pretreated with IL-2/JES6 at different time points after LPS challenge. Levels of TNF-α, IFN-γ, IL-1β, IL-12, and IL-6 were much higher in sera of C56BL/6 mice pretreated with IL-2/JES6 in comparison with controls, despite the fact that control mice were challenged with 10-times higher doses of LPS. Contrary to this, levels of the above mentioned cytokines were comparable in sera of control and IL-2/JES6 pretreated BALB/c mice (*Figure 4A*). Furthermore, BALB/c mice pretreated with IL-2/JES6 showed much milder sensitization to LPS in term of hypothermia and mortality than C56BL/6 mice (*Figure 4B*). These data show that the endogenous production of IFN-γ upon IL-2/JES6 administration probably plays a crucial role in sensitization to LPS. To confirm a key role of IFN-γ directly, we injected C56BL/6 mice with anti-IFN-γ mAb prior to IL-2/JES6 pretreatment

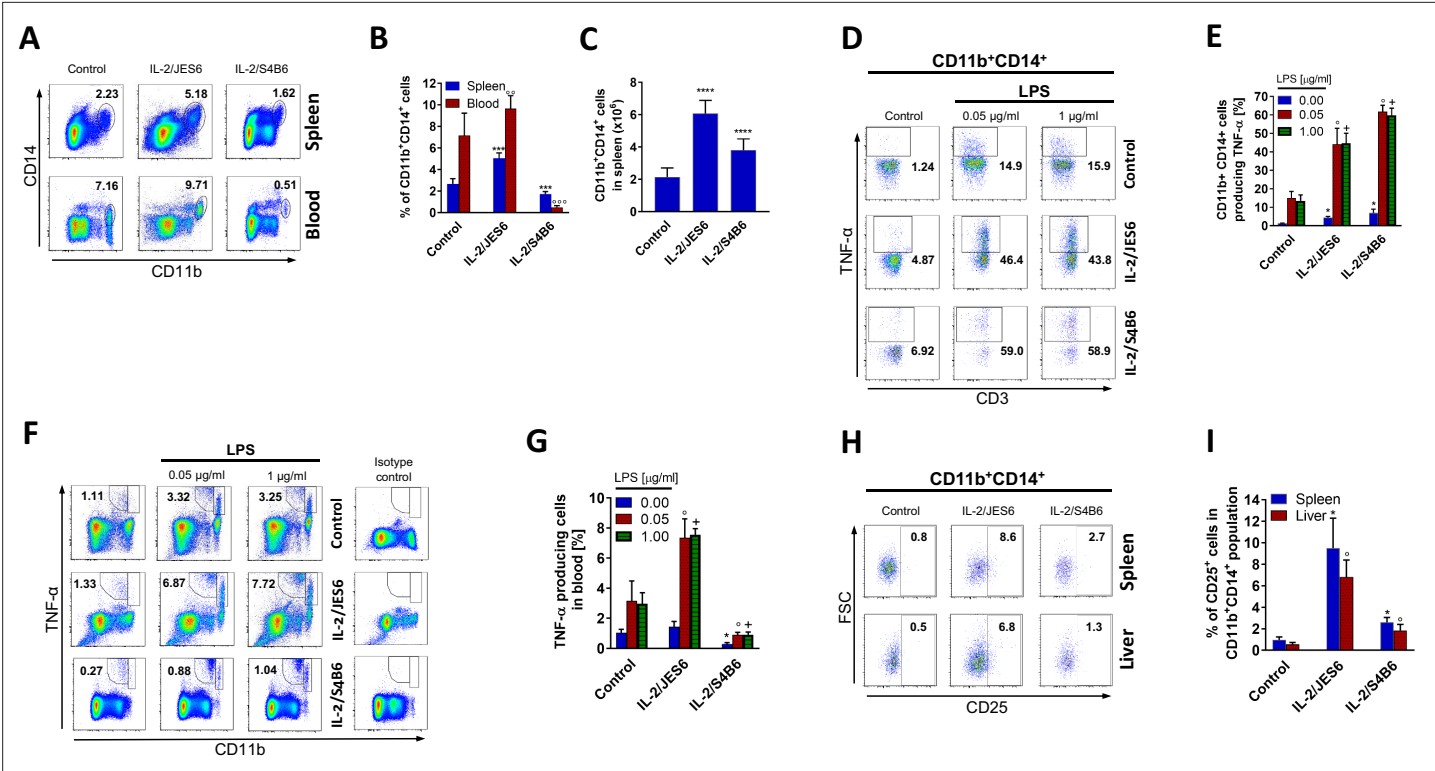

**Figure 3.** IL-2/JES6 increases counts of CD11b$^+$CD14$^+$ cells and their responsiveness to LPS in terms of TNF-α production. C56BL/6 mice were treated with IL-2/JES6 or IL-2/S4B6 as shown in *Figure 1A*. The relative number of CD11b$^+$CD14$^+$ cells was determined by flow cytometry in blood and spleen 2 d after the last dose of IL-2 complexes. Dot plots showing one representative mouse (**A**) and a bar graph showing the mean ± SD in the experimental groups (**B**) are shown. (**C**) Absolute numbers of CD11b$^+$CD14$^+$ cells in the spleen of mice from the same experiment as shown in A and B. C56BL/6 mice were treated with IL-2/JES6 or IL-2/S4B6 as shown in *Figure 1A*. Spleen cells were cultivated ex vivo with LPS for 2 h and TNF-α production in CD11b$^+$CD14$^+$ cells was determined by flow cytometry. Dot plots showing one representative mouse (**D**) and a bar graph showing the mean ± SD in experimental groups (**E**) are shown. A similar experiment to the one shown in D and E was done using the blood of C56BL/6 mice treated with IL-2/JES6 or IL-2/S4B6. TNF-α production in CD11b$^+$ cells is shown in one representative mouse (**F**) and in a bar graph showing the mean ± SD in experimental groups (**G**). (**H**) C56BL/6 mice were treated with IL-2/JES6 or IL-2/S4B6 as shown in *Figure 1A*. Flow cytometry analysis of spleen and liver cells was used to evaluate CD25 expression in CD11b$^+$CD14$^+$ cells. Dot plots showing one representative mouse (**H**) and a bar graph showing mean ± SD in experimental groups (**I**) are presented. All experiments were done at least twice with similar results; n = 3–10 technical replicates. Data were analysed using an unpaired two-tailed Student's t-test. Significant differences to control are shown (*,°,+ p ≤ 0.05; °° p ≤ 0.01; ***, °°° p ≤ 0.001).

The online version of this article includes the following source data and figure supplement(s) for figure 3:

**Source data 1.** Source data for *Figure 3*, panels B, C, E, G, and I.

**Figure supplement 1.** IL-2/JES6 expands various subsets of myeloid cells in the spleen.

**Figure supplement 1—source data 1.** Source data for *Figure 3—figure supplement 1*, panels A-G.

**Figure supplement 2.** IL-2/JES6 increase MHC II expression in monocyte/macrophage population in the spleen.

**Figure supplement 2—source data 1.** Source data for *Figure 3—figure supplement 2*, panels A-D.

**Figure supplement 3.** IL-2/JES6 promote proliferation and expansion of myeloid cells in dose-dependent manner in the spleen.

**Figure supplement 3—source data 1.** Source data for *Figure 3—figure supplement 3*, panel A.

**Figure supplement 4.** The expression of TLR4 in splenocytes of C57BL/6 mice is not affected by the treatment with IL-2/JES6 or IL-2/S4B6.

**Figure supplement 4—source data 1.** Source data for *Figure 3—figure supplement 4*.

**Figure supplement 5.** IL-2/JES6 do not increase the level of TLR4 in myeloid cells in the spleen.

**Figure supplement 5—source data 1.** Source data for *Figure 3—figure supplement 5*, panels B, D and F.

**Figure supplement 6.** IL-2/JES6 increase counts of CD45+ and CD11b + CD14+ cells in the liver.

**Figure supplement 6—source data 1.** Source data for *Figure 3—figure supplement 6*, panels A-C.

**Figure supplement 7.** IL-2/JES6 increase counts of CD45+ and CD11b + CD14+ cells in the lungs.

**Figure supplement 7—source data 1.** Source data for *Figure 3—figure supplement 7*, panels A-C.

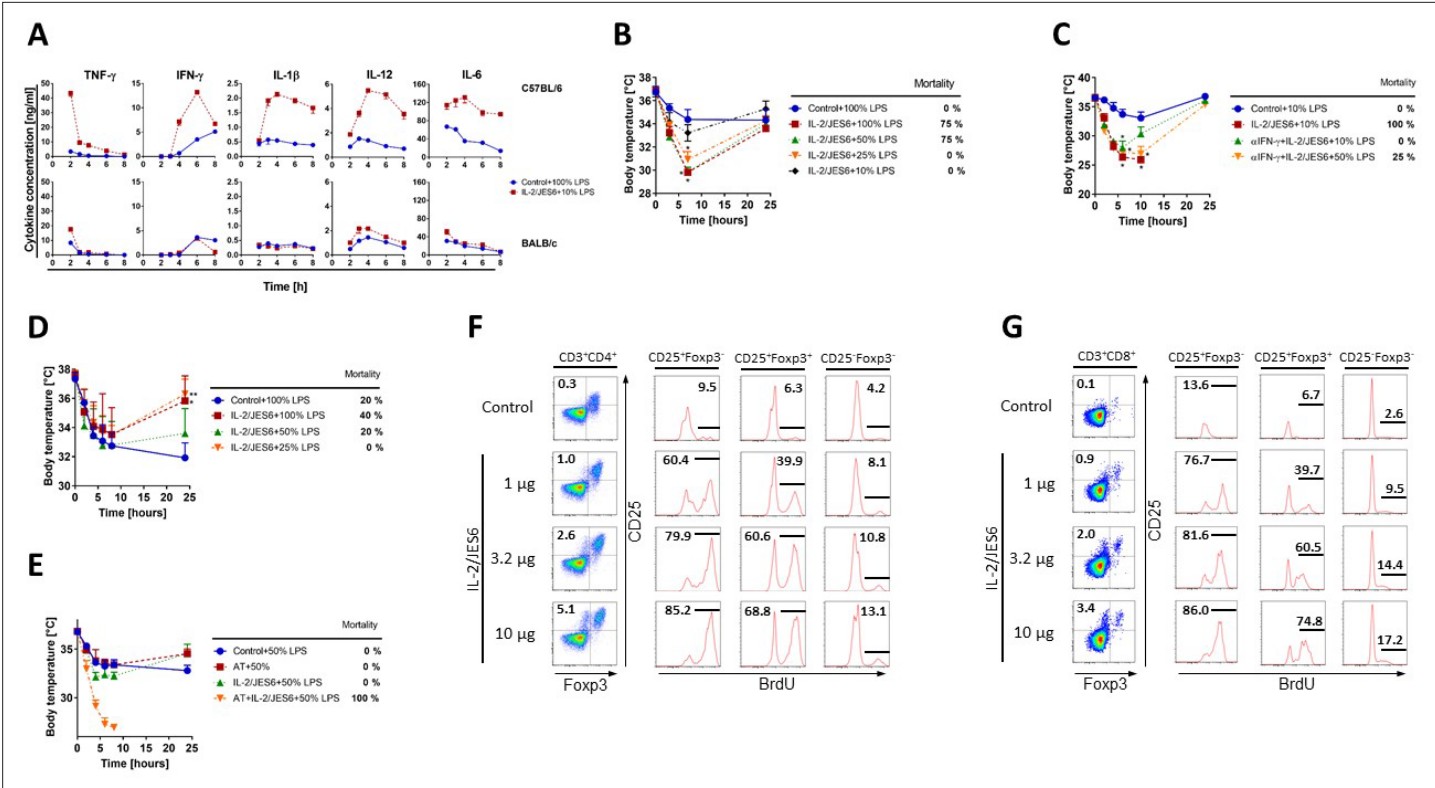

**Figure 4.** Sensitization to LPS via IL-2/JES6 requires endogenous IFN-γ production and is T cell-dependent. (**A**) C56BL/6 and BALB/c mice were treated with IL-2/JES6 and challenged with LPS (10% of MNLD) as shown in *Figure 1A*. Control mice were treated with PBS and challenged with LPS (100% of MNLD). Mice were euthanized at selected time points and their individual sera were collected. Concentrations of cytokines in the serum were determined by ELISA. Each experimental point represents the mean of two mice ± SD (n = 2 technical replicates). (**B**) BALB/c mice were treated with IL-2/JES6 and challenged with titrated doses of LPS as shown in *Figure 1A*. (**C**) C56BL/6 mice were treated with IL-2/JES6 and challenged with LPS (10 or 50% of MNLD) as shown in *Figure 1A*. Some mice were also injected with anti-IFN-γ mAb (αIFN-γ; 250 µg/mice i.p.) 4 h prior to the first injection of IL-2/JES6. (**D**) Nu/Nu mice were treated with IL-2/JES6 and challenged with titrated doses of LPS as shown in *Figure 1A*. (**E**) Two groups of Nu/Nu mice were adoptively transferred (AT) with 2 × 10^6 CD4+CD25+ T cells from C56BL/6 mice pretreated with IL-2/JES6 as shown in *Figure 1A*. One group of AT mice and one group of normal Nu/Nu mice were treated with IL-2/JES6 as shown in *Figure 1A*. All groups including Nu/Nu mice without AT and IL-2/JES6 treatment (Control) were challenged with LPS (50% of MNLD). C56BL/6 mice were injected with one titrated dose of IL-2/JES6 or with PBS (Control). Mice were injected i.p. with BrdU 4 h after injection of IL-2/JES6 and put on drinking water with BrdU. Mice were euthanized 48 h after the injection of IL-2/JES6 and their CD4+ and CD8+ T cells (F and G, respectively) from the spleen were analysed by flow cytometry. One representative mouse out of two for each condition is shown. All experiments were done at least twice with similar results; n = 2–5 technical replicates (**B–G**). Data were analysed using an unpaired two-tailed Student's t-test. Significant differences to control are shown (* p ≤ 0.05; ** p ≤ 0.01).

The online version of this article includes the following source data and figure supplement(s) for figure 4:

**Source data 1.** Source data for *Figure 4*, panels A-E.

**Figure supplement 1.** TLR4 signalling in T cells is dispensable for inducing LPS hypersensitivity by IL-2/JES6.

**Figure supplement 1—source data 1.** Source data for *Figure 4—figure supplement 1*.

**Figure supplement 2.** Anti-IFN-γ mAb protects from sensitization to LPS more effectively when administered before treatment with IL-2/JES6.

**Figure supplement 2—source data 1.** Source data for *Figure 4—figure supplement 2*, panel B.

and LPS challenge. Indeed, anti-IFN-γ mAb markedly diminished sensitization to LPS by IL-2/JES6 (*Figure 4C*). Hence, we conclude that endogenous production of IFN-γ is a prerequisite for strong sensitization to LPS via IL-2/JES6.

Next, we asked which cells are key IFN-γ producers upon IL-2/JES6 administration. Since T cells are the most prominent IFN-γ producers, we decided to test sensitization to LPS by IL-2/JES6 in athymic Nu/Nu mice lacking T cells. We found that IL-2/JES6 almost do not sensitize Nu/Nu mice to LPS (*Figure 4D*). On the other hand, Nu/Nu mice with adoptively transferred CD4+CD25+ T cells from C56BL/6 mice injected with IL-2/JES6 showed very high sensitization to LPS upon IL-2/JES6 treatment (*Figure 4E*). These data show that IL-2/JES6 is able to induce IFN-γ production in T cells even in

the absence of TCR signal except that provided by self-MHC molecules. TLR4 expression on T cells seems to be irrelevant for their ability to sensitize the mice to LPS since $Rag1^{-/-}$ mice with adoptively transferred T cells from $Myd88^{-/-}$ mice, that is with severely impaired TLR4 downstream signaling, showed profound LPS sensitivity upon treatment with IL-2/JES6 (*Figure 4—figure supplement 1*). The key question is therefore, which T cell subset(s) produces IFN-γ upon IL-2/JES6 treatment? The ability to utilize IL-2/JES6 strictly requires CD25 expression and the only T cells expressing CD25 in naïve unprimed mice are $T_{reg}$ cells. However, $T_{reg}$ cells are considered as a T cell subset with no ability to produce IFN-γ under normal conditions. Nevertheless, we found that administration of IL-2/JES6 potently expanded CD25+Foxp3- T cells in both CD4+ and CD8+ subsets and that these cells proliferate more vigorously in response to IL-2/JES6 than $T_{reg}$ cells (*Figure 4F and G*).

## IL-2/JES6 drive expansion of CD25+Foxp3- T cells producing IFN-γ in lymphoid as well as non-lymphoid tissues

We focused on CD25+Foxp3-CD4+ and CD8+ T cells robustly expanded in the spleen of mice treated with IL-2/JES6. Since these cells resemble by their phenotype activated T cells, we presumed that these cells could be a key producers of IFN-γ in IL-2/JES6 treated mice causing the increased sensitivity to LPS. Thus, we decided to investigate whether these cells were expanded also in other organs except of spleen and whether they produced IFN-γ. Interestingly, we found that treatment of C56BL/6 mice with IL-2/JES6 led to the induction of these cells in both lymphoid tissues (spleen) as well as in non-lymphoid tissues (liver and lungs; *Figure 5A and B*). Relative counts of CD25+Foxp3-CD4+ and CD8+ T cells were particularly high in the liver where they typically represented about 6% and 10% of CD4+ and CD8+ T cells, respectively. We asked whether these CD25+Foxp3-CD4+ and CD8+ T cells are able to produce IFN-γ upon IL-2/JES6 administration in vivo. To answer this, C56BL/6 mice were treated with IL-2/JES6 as in *Figure 1A* and brefeldin A was injected 2 h after the last dose of IL-2/JES6 to enable the detection of IFN-γ production in various T cell subsets intracellularly. We proved that CD25+Foxp3-CD4+ and CD8+ T cells in spleen, liver, and lungs produced IFN-γ (*Figure 5C and D*). Again, relative counts of these cells producing IFN-γ were especially high in the liver and generally higher in the CD8+ subset in comparison to the CD4+ subset. Notably, $T_{reg}$ cells in the liver of C56BL/6 mice treated with IL-2/JES6 also produced IFN-γ showing that $T_{reg}$ cells may surprisingly participate in sensitization to LPS. IL-2/S4B6 almost did not induce CD25+Foxp3-CD4+ and CD8+ T cells in any tissue studied. This agrees with the previous finding that IL-2/S4B6 sensitize to LPS only very decently.

## Production of IFN-γ by antigen-activated or IL-2/JES6-stimulated T cells drives LPS hyperreactivity

Previous experiments showed that IL-2/JES6 give rise to CD25+Foxp3- T cells producing IFN-γ in various tissues which in turn leads to LPS hyperreactivity. We asked whether antigen-activated T cells, when present in sufficient numbers, were also able to induce LPS hyperreactivity since they also produce IFN-γ. Thus, we adoptively transferred CD8+ OT-I and CD4+ OT-II T cells into C56BL/6 mice and activated them with respective ovalbumin-derived peptides plus polyI:C. CD8+ OT-I and CD4+ OT-II T cells significantly expanded and altogether made up 12–13% of all T cells in the spleen on day 3 post priming (*Figure 6A and B*). C56BL/6 mice with primed adoptively transferred T cells showed significant sensitivity to LPS in comparison to C56BL/6 mice with unprimed adoptively transferred T cells (*Figure 6C*). Importantly, injection of anti-IFN-γ mAb abrogated sensitization to LPS in C56BL/6 mice with primed adoptively transferred T cells. C56BL/6 mice without adoptively transferred CD8+ OT-I and CD4+ OT-II T cells and immunized with OVA plus polyI:C showed no increased sensitivity to LPS (*Figure 6—figure supplement 1*).

Finally, we used IFN-γ deficient mice (IFN-γ$^{-/-}$) to prove that IFN-γ is the key factor mediating LPS hyperreactivity upon IL-2/JES6 administration. IFN-γ$^{-/-}$ mice pretreated with IL-2/JES6 showed sensitivity to LPS comparable to that of those not pretreated. IFN-γ sufficient C56BL/6 mice pretreated with IL-2/JES6 showed dramatically increased sensitivity to LPS (*Figure 6D and E*). Altogether, we conclude that IL-2/JES6 give rise to CD25+Foxp3-CD4+ and CD8+ T cells producing IFN-γ in various tissues, particularly in liver. IL-2/JES6 also expand LPS responsive CD11b+CD14+ myeloid cells and increase expression of CD25 in these cells. IFN-γ later acts on these myeloid cells and increases their responsiveness to LPS (*Figure 6F*).

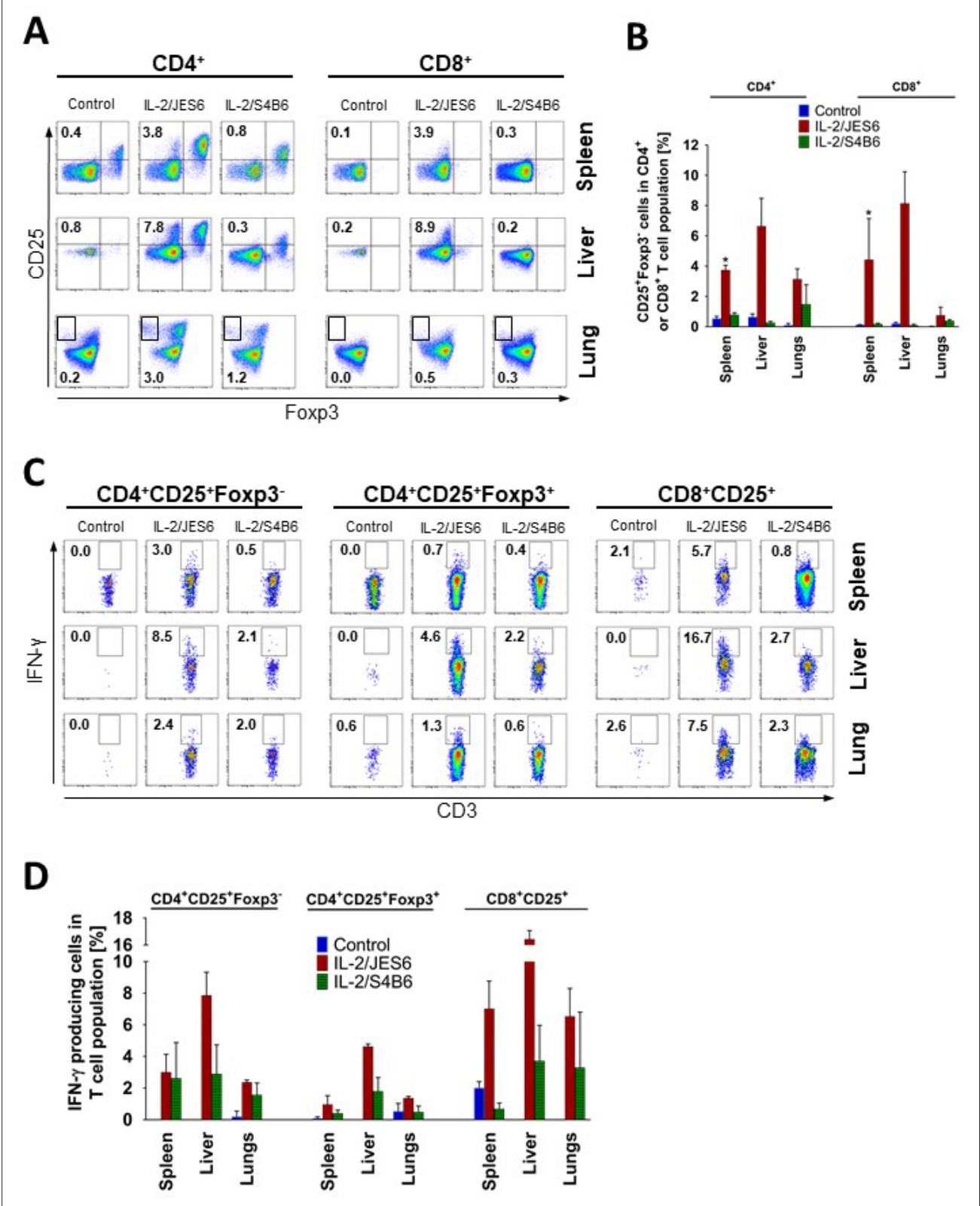

**Figure 5.** IL-2/JES6 expands CD25+Foxp3-T cells in both CD4+ and CD8+ subsets which produces IFN-γ in various tissues. (**A**) C56BL/6 mice were treated with IL-2/JES6 or IL-2/S4B6 as shown in *Figure 1A*. Mice were i.p. injected with 150 µg/mouse of brefeldin A 2 h after the last dose of IL-2/JES6 and euthanized 12 h after the injection of brefeldin A. CD4+ and CD8+ T cells from spleen, liver, and lung were analysed by flow cytometry for CD25 and Foxp3 expression. Dot plots showing one representative mouse (**A**) and a bar graph showing mean ± SD in experimental groups (**B**) are presented.

*Figure 5 continued on next page*

*Figure 5 continued*

Intracellular production of IFN-γ in different subpopulations of T cells as shown in A, was determined. Dot plots showing one representative mouse (**C**) and a bar graph showing mean ± SD in experimental groups (**D**) are presented. Analysis of T cells from spleen and liver was carried out in the same experiment, but analysis of T cells from the lungs was done in a separate experiment. All experiments were done at least twice with similar results; n = 3 technical replicates. Data were analysed using an unpaired two-tailed Student's t-test. Significant differences to control are shown (* $p \leq 0.05$).

The online version of this article includes the following source data for figure 5:

**Source data 1.** Source data for *Figure 5*, panels B and D.

## Discussion

It has previously been shown that IL-2/JES6 immunocomplexes potently expand $T_{reg}$ cells in vivo and that they can be used for treatment of various autoimmune diseases (*Liu et al., 2011*; *Webster et al., 2009*; *Wilson et al., 2008*). However, our results clearly demonstrate that these immunocomplexes also dramatically increase sensitivity to LPS. The key mechanism is the production of IFN-γ since administration of anti-IFN-γ mAb as well as the use of IFN-γ$^{-/-}$ mice abrogates the sensitization. This resembles the so-called Schwartzman-like shock reaction, an experimental model of lethal systemic inflammatory reaction induced by LPS injection. It is induced by two consecutive, rather low doses of LPS (preparatory and provocative ones) and occurrence of the reaction requires very exact dosage and careful timing (*Heremans et al., 1990*). The first LPS injection is given into the footpad, followed after 24 h by an i.v. dose. Administration of anti-IFN-γ mAb was found to completely prevent the reaction. Thus, a preparatory dose of LPS induces production of IFN-γ which consequently sensitizes immune cells to be hyperresponsive to a provocative dose of LPS. Contrary to the Schwartzman-like shock reaction, IL-2/JES6 is responsible for inducing the production of IFN-γ and for sensitization to LPS in our experimental system. Administration of anti-IFN-γ mAb before IL-2/JES6 treatment could, however, have two effects: it could prevent sensitization of immune cells to LPS or it could protect against the toxic effect of a massive production of IFN-γ upon LPS injection since IgG has a relatively long half-life (*Vieira and Rajewsky, 1988*). Thus, we compared the effect of anti-IFN-γ mAb administered either before pretreatment with IL-2/JES6 or after it, that is, shortly before LPS challenge (4 h). Anti-IFN-γ mAb had higher protective effect when injected before IL-2/JES6 pretreatment (*Figure 4—figure supplement 2*) further confirming the key role of IL-2/JES6-induced IFN-γ production for sensitization to LPS.

The key question was: which cells produce IFN-γ upon treatment with IL-2/JES6? IFN-γ is normally produced by effector T cells after their activation and expansion. It requires a TCR signal and is significantly augmented by the presence of IL-12 and IL-18 (*Berg et al., 2002*; *Li et al., 2005*; *Nakanishi, 2018*). However, IL-2 was shown to be also able to promote IFN-γ production, but usually in activated T cells, that is, upon TCR signaling (*Boyman and Sprent, 2012*; *Cousens et al., 1995*). Here, we show that a strong sustained IL-2 signal provided by IL-2/JES6 is able to induce production of IFN-γ in T cells in vivo even in the absence of a TCR signal, except that provided by self-MHC molecules. Contact with self-MHC class I and II molecules (low affinity interaction) is known to play an important role in homeostasis of naïve CD8$^+$ and CD4$^+$ T cells, respectively (*Kieper et al., 2004*; *Sprent and Surh, 2011*), but should not endow these cells with the capacity to express effector functions like IFN-γ production (*Kieper et al., 2004*). We saw the highest relative counts of IFN-γ-producing T cells in CD25$^+$Foxp$^-$CD4$^+$ and CD8$^+$ populations. Nevertheless, CD4$^+$CD25$^+$Foxp$^+$ $T_{reg}$ cells also produced IFN-γ in IL-2/JES6-treated mice, particularly in liver (*Figure 5C and D*). One simple explanation could be that the strong sustained IL-2 signal provided by IL-2/JES6 also induces production of IFN-γ in $T_{reg}$ cells. However, we speculate that the IFN-γ produced by CD25$^+$Foxp$^-$CD4$^+$ and CD8$^+$ cells may be a trigger for IFN-γ production in $T_{reg}$ cells since it was shown previously that $T_{reg}$ cells can be converted into IFN-γ-producing cells, especially in a strongly pro-inflammatory milieu (*Koenecke et al., 2012*) and/or in the presence of IFN-γ (*Wood and Sawitzki, 2006*). On the other hand, $T_{reg}$ cells seems to be dispensable for sensitization to LPS via IL-2/JES6 as administration of anti-CD4 mAb had no effect (*Figure 2D*). This shows that CD8$^+$ T cells can substitute CD4$^+$ T cells in the process of sensitization, however, complete absence of T cells abolishes it (*Figure 4D*).

It has been shown that treatment with nitrite (NO$_2^-$), an important biological NO reservoir in vasculature and tissues, significantly attenuates hypothermia, tissue infarction and mortality in a mouse shock model induced by a lethal TNF-α or LPS challenge (*Cauwels et al., 2009*). We therefore asked

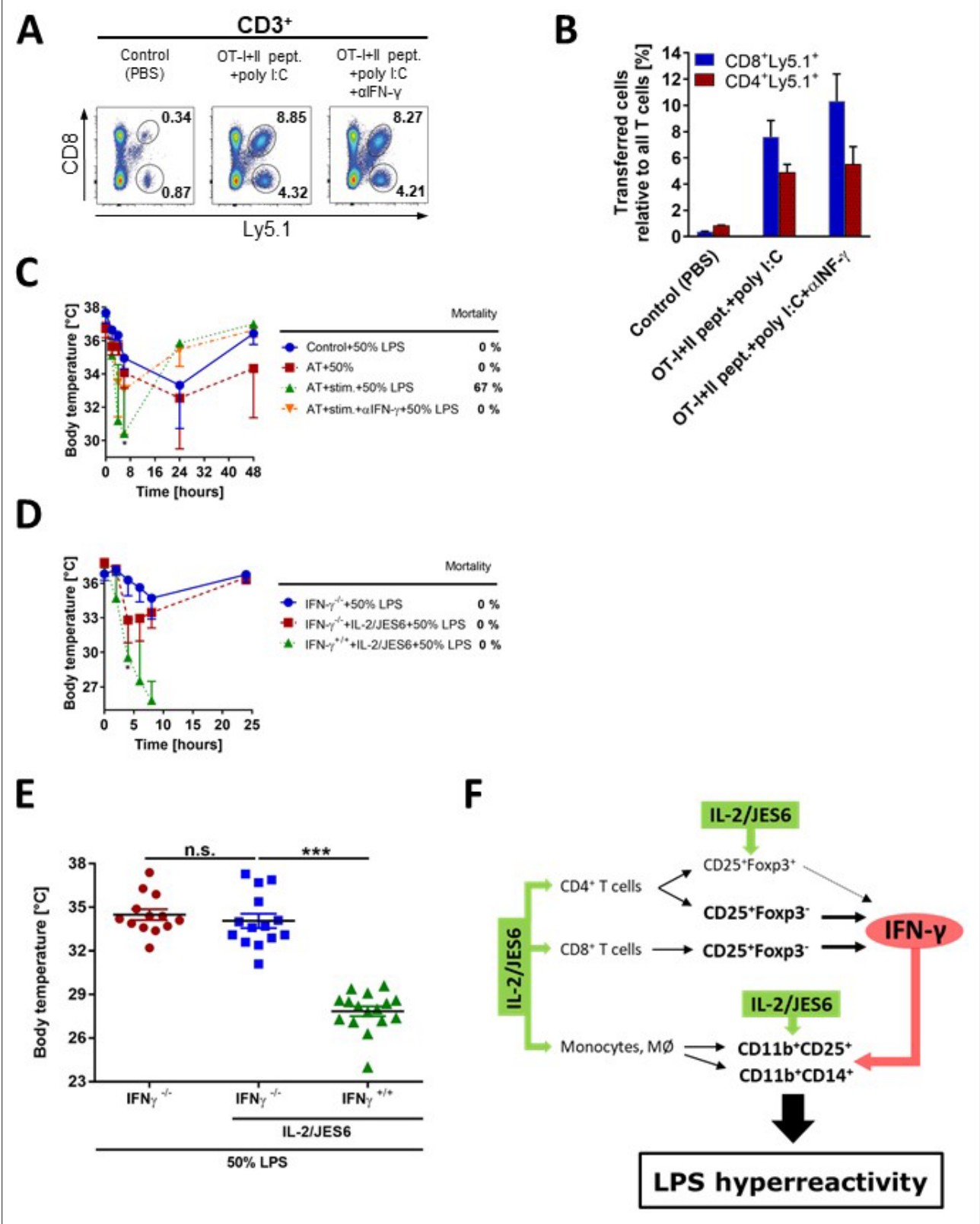

**Figure 6.** IFN-γ produced by activated antigen-specific T cells or by T cells stimulated with IL-2/JES6, is a crucial mediator of LPS hyperreactivity. Purified CD8+ and CD4+ T cells from OT-I/Ly5.1 and OT-II/Ly5.1 mice (2 and 4 × 10⁶ /mouse respectively), were adoptively transferred (AT) to C56BL/6 mice. One day later, mice were i.p. injected with PBS (Control), with both OT-I and OT-II specific peptides plus polyI:C (75 µg/mice) or with the latter plus αIFN-γ mAb (250 µg). Expansion of AT T cells was determined 3 d post stimulation by flow cytometry. Dot plots showing one representative

*Figure 6 continued on next page*

*Figure 6 continued*

mouse (**A**) out of two mice analysed by flow cytometry and a bar graph showing mean ± SD (**B**) are presented. (**C**) Three groups of C56BL/6 mice described above and one control group (no AT) were challenged with LPS (50% of MNLD). (**D**) IFN-γ$^{-/-}$ and normal C56BL/6 mice (IFN-γ$^{+/+}$) were treated with IL-2/JES6 and challenged with LPS (50% of MNLD) as shown in *Figure 1A*. IFN-γ$^{-/-}$ C56BL/6 mice challenged with the same dose of LPS were used as the control. (**E**) Data pooled from three independent experiments (n = 13–16 technical replicates) described in D showing body temperature of mice 8 h after LPS challenge. (**F**) Scheme showing the proposed mechanism of how IL-2/JES6 induces LPS hyperreactivity. Experiments A-D were done at least twice with similar results; n = 2–6 technical replicates. Data were analysed using unpaired two-tailed Student's t-test. Significant differences to control are shown (* $p \leq 0.05$; *** $p \leq 0.001$).

The online version of this article includes the following source data and figure supplement(s) for figure 6:

**Source data 1.** Source data for *Figure 6*, panels B-E.

**Figure supplement 1.** Immunization with ovalbumin plus poly I:C does not increase sensitivity to LPS.

**Figure supplement 1—source data 1.** Source data for *Figure 6—figure supplement 1*.

whether nitrite could protect, at least to some extent, IL-2/JES6 pretreated mice challenged with LPS, since we observed very high serum levels of TNF-α in these mice (*Figure 1E*). Indeed, nitrite was considerably effective in lowering hypothermia and mortality (*Figure 1—figure supplement 1*) showing that a massive production of TNF-α is a dominant pathogenic factor responsible for rapid onset of hypothermia in IL-2/JES6 sensitized mice challenged with LPS. Pathogenic effects that occur later on are mediated by other cytokines, particularly by IFN-γ. This is supported by the fact that serum levels of TNF-α peak very early (1–2 h) post LPS challenge while serum levels of IFN-γ peak around 6 h (*Figure 4A*).

Treatment with IL-2/JES6 not only affected T cell populations but to our surprise also affected some myeloid cell populations, particularly CD11b$^+$CD14$^+$ cells (monocytes/macrophages). Although IL-2 is a pleiotropic cytokine, it has stimulatory activity mostly for cells of lymphoid origin, especially for T$_{reg}$, recently activated T, memory CD8$^+$ T, and NK cells (*Boyman et al., 2010*; *Malek, 2008*; *Sharma and Das, 2018*; *Yu et al., 2000*). Increased counts of CD11b$^+$CD14$^+$ cells in response to IL-2/JES6 are thus intriguing. We tested the hypothesis that T cells upon stimulation with IL-2/JES6 do not produce GM-CSF but no production of this cytokine was found. However, there is a report showing that clonogenic common lymphoid progenitor (CLP), a bone marrow-resident cell that gives rise exclusively to lymphocytes, can be redirected to the myeloid lineage by stimulation through an exogenously expressed IL-2 receptor (*Kondo et al., 2000*). Authors suggest that CLPs and pro-T cells may have a latent granulocyte/monocyte lineage differentiation program that can be initiated by signaling through the reconstituted IL-2 receptor. We hypothesize that a strong IL-2 signal provided by IL-2/JES6 may lead to a similar effect. Nevertheless, this hypothesis requires that at least a fraction of CLPs express complete trimeric IL-2R at some stage of their development. Even low expression levels should be sufficient since it has been previously shown that IL-2/JES6 feeds back onto CD25 expression (*Spangler et al., 2015*).

An interesting feature of IL-2/JES6 treatment is the increased counts of CD45$^+$ cells (i.e. any haematopoietic cells, except erythrocytes) in the liver (*Figure 3—figure supplement 6*). This increase was found both in the T cell population (see later) as well as in CD11b$^+$CD14$^+$ cells (*Figure 3—figure supplement 6B* and C). A decent increase in counts of CD11b$^+$CD14$^+$ cells was also found in the lungs (*Figure 3—figure supplement 7*). We believe that accumulation of these cells in liver and lungs might significantly contribute to the pathological effects seen after LPS challenge in IL-2/JES6 pretreated mice, since these cells are likely to be sensitized to LPS via IFN-γ produced by CD25$^+$ T cells. This is not the case with IL-2/S4B6 pretreatment as there are much fewer IFN-γ-producing CD25$^+$ T cells (*Figure 5*). Further, treatment with IL-2/JES6 per se induces more severe lung oedema in comparison to treatment with IL-2/S4B6, which could contribute to the morbidity and mortality after LPS challenge to some extent (*Figure 1—figure supplement 2*).

We injected both IL-2/JES6 and LPS into mice via i.p. injections. There is a population of macrophages in the peritoneal cavity (*Cassado et al., 2015*) and therefore we asked whether these cells might contribute to the LPS sensitivity. However, adoptive transfer of peritoneal exudate cells from mice i.p. injected with IL-2/JES6 did not increase sensitivity to LPS in adoptively transferred mice. Moreover, i.v. administration of IL-2/JES6 leads to the same level of LPS sensitivity in comparison to i.p. administration. Therefore, we can rule out the idea that peritoneal macrophages play an important role.

We found significantly increased counts of T cells in the liver of mice treated with IL-2/JES6 or IL-2/S4B6 (*Figure 5A*). We do not know the mechanism of T cell accumulation in the liver upon treatment with either IL-2 complexes, however, we can speculate that IL-2 complexes ($M_w$ ~190 kDa) are easily accessible in the liver interstitium due to the massive fenestration of capillaries in this organ (*DeLeve, 2015*). It seems that the liver and secondary lymphoid organs are the sites where increased numbers of CD11b⁺CD14⁺ and IFN-γ-producing T cells colocalize upon IL-2/JES6 treatment, and this feature governs LPS hyperresponsiveness.

# Materials and methods

**Key resources table**

| Reagent type (species) or resource | Designation | Source or reference | Identifiers | Additional information |
|---|---|---|---|---|
| Antibody | Anti-mouse CD3-eFluor 450 (clone: 17A2; rat monoclonal) | eBioscience | Cat#: 48-0032-82; RRID:AB_1272193 | Flow Cytometry: (dilution 1:30) |
| Antibody | Anti-mouse CD8a-PerCP-Cyanine5.5 (clone: 53–6.7; rat monoclonal) | eBioscience | Cat#: 45-0081-82; RRID:AB_1107004 | Flow Cytometry: (dilution 1:100) |
| Antibody | Anti-mouse CD11b-Alexa Fluor 700 (clone: M1/70; rat monoclonal) | eBioscience | Cat#: 56-0112-82; RRID:AB_1107004 | Flow Cytometry: (dilution 1:300) |
| Antibody | Anti-mouse CD11b-eFluor450 (clone: M1/70; rat monoclonal) | eBioscience | Cat#: 48-0112-82; RRID:AB_1582236 | Flow Cytometry: (dilution 1:500) |
| Antibody | Anti-mouse CD14-FITC (clone: Sa2-8; rat monoclonal) | eBioscience | Cat#: 11-0141-85; RRID:AB_464950 | Flow Cytometry: (dilution 1:20) |
| Antibody | Anti-mouse CD25-APC (clone: PC61.5; rat monoclonal) | eBioscience | Cat#: 17-0251-82; RRID:AB_469366 | Flow Cytometry: (dilution 1:500) |
| Antibody | Anti-mouse CD25-eFluor 450 (clone: PC61.5; rat monoclonal) | eBioscience | Cat#: 48-0251-82; RRID:AB_10671550 | Flow Cytometry: (dilution 1:400) |
| Antibody | Anti-mouse CD45.1-APC (clone: A20; mouse monoclonal) | eBioscience | Cat#: 17-0453-82; RRID:AB_469398 | Flow Cytometry: (dilution 1:200) |
| Antibody | Anti-mouse Ly-6C-APC-eFluor 780 (clone: HK1.4; rat monoclonal) | eBioscience | Cat#: 47-5932-82; RRID:AB_2573992 | Flow Cytometry: (dilution 1:200) |
| Antibody | Anti-mouse MHC Class II (I-A/I-E)-FITC (clone: M5/114.15.2; rat monoclonal) | eBioscience | Cat#: 11-5321-82; RRID:AB_465232 | Flow Cytometry: (dilution 1:300) |
| Antibody | Anti-mouse CD45.1-eFluor 450 (clone: A20; mouse monoclonal) | eBioscience | Cat#: 48-0453-82; RRID:AB_1272189 | Flow Cytometry: (dilution 1:200) |
| Antibody | Anti-mouse CD45/B220-Horizont V500 (clone: RA3-6B2; rat monoclonal) | BD Biosciences | Cat#: 561226; RRID:AB_10563910 | Flow Cytometry: (dilution 1:100) |
| Antibody | Anti-mouse CD3e-Horizont V500 (clone: 500A2; Syrian hamster monoclonal) | BD Biosciences | Cat#: 560771; RRID:AB_1937314 | Flow Cytometry: (dilution 1:30) |
| Antibody | Anti-mouse CD4-Horizont V500 (clone: RM4.5; rat monoclonal) | BD Biosciences | Cat#: 560782; RRID:AB_1937327 | Flow Cytometry: (dilution 1:400) |
| Antibody | Anti-mouse CD4-PerCP (clone: RM4-5; rat monoclonal) | BD Biosciences | Cat#: 553052; RRID:AB_394587 | Flow Cytometry: (dilution 1:200) |
| Antibody | Anti-mouse CD8a-V500 (clone: 53–6.7; rat monoclonal) | BD Biosciences | Cat#: 560776; RRID:AB_1937317 | Flow Cytometry: (dilution 1:80) |

*Continued on next page*

*Continued*

| Reagent type (species) or resource | Designation | Source or reference | Identifiers | Additional information |
|---|---|---|---|---|
| Antibody | Anti-mouse Ly-6G-Alexa Fluor 700 (clone: 1A8; rat monoclonal) | BD Biosciences | Cat#: 561236; RRID:AB_10611860 | Flow Cytometry: (dilution 1:80) |
| Antibody | Anti-mouse Siglec-F-PE (clone: E50-2440; rat monoclonal) | BD Biosciences | Cat#: 562068; RRID:AB_394341 | Flow Cytometry: (dilution 1:100) |
| Antibody | Anti-mouse TLR4-APC (clone: SA15-21; rat monoclonal) | BioLegend | Cat#: 145406; RRID:AB_2562503 | Flow Cytometry: (dilution 1:300) |
| Antibody | Anti-mouse TNF alpha-PE (clone: MP6-XT22; rat monoclonal) | eBioscience | Cat#: 12-7321-82; RRID:AB_466199 | Flow Cytometry: (dilution 0.35 µL/test) |
| Antibody | Anti-FOXP3-PE (clone: FJK/16 s; rat monoclonal) | eBioscience | Cat#: 12-5773-82; RRID:AB_465936 | Flow Cytometry: (dilution 1 µL/test) |
| Antibody | Anti-mouse IFN gamma-PE (clone: XMG1.2; rat monoclonal) | eBioscience | Cat#: 12-7311-82; RRID:AB_466193 | Flow Cytometry: (dilution 0.35 µL/test) |
| Antibody | Anti-mouse CD16/32 (clone: 03; rat monoclonal) | eBioscience | Cat#: 14-0161-86; RRID:AB_467135 | Flow Cytometry: (dilution 0.5 µg/test) |
| Antibody | Anti-mouse IFN gamma (clone: XMG1.2; rat monoclonal) | BioXCell | Cat#: BE0055; RRID:AB_1107694 | In vivo IFN gamma neutralization (dose: 250 µg/mouse) |
| Antibody | Anti-mouse IL-2 (clone: JES6-1A12; rat monoclonal) | BioXCell | Cat#: BE0043; RRID:AB_1107702 | In vivo IL-2 receptor stimulation as a complex with IL-2 (dose: 1.5 µg /mouse) |
| Antibody | Anti-mouse IL-2 (clone: S4B6-1; rat monoclonal) | BioXCell | Cat#: BE0043-1; RRID:AB_1107705 | In vivo IL-2 receptor stimulation as a complex with IL-2 (dose: 1.5 µg /mouse) |
| Antibody | Anti-mouse CD4 (clone: GK1.5; rat monoclonal) | BioXCell | Cat#: BE0003-1; RRID:AB_1107636 | In vivo CD4+ T cell depletion (dose: 50 µg / mouse) |
| Antibody | Anti-mouse CD25 (clone: PC61.5; rat monoclonal) | BioXCell | Cat#: BE0012; RRID:AB_1107619 | In vivo regulatory T cell depletion (dose: 50 µg /mouse) |
| Sequence-based Reagent | *Casc3*for | Sigma-Aldrich | | 5′-TTCGAGGTGTGCCTAACCA-3′ |
| Sequence-based Reagent | *Casc3*rev | Sigma-Aldrich | | 5′-GCTTAGCTCGACCACTCTGG-3′ |
| Sequence-based Reagent | *H6pd*for | Sigma-Aldrich | | 5′-GGATTGTGTTTAAGAATCGGG-3′ |
| Sequence-based Reagent | *H6pd*rev | Sigma-Aldrich | | 5′-AGTAGGCGTCTTGCTC-3′ |
| Sequence-based Reagent | *Tlr4*for | Sigma-Aldrich | | 5′-GATCATGGCACTGTTCTTCTC-3′ |
| Sequence-based Reagent | *Tlr4*rev | Sigma-Aldrich | | 5′-CACACCTGGATAAATCCAGC-3′ |
| Strain, Strain Background (*Salmonella Typhymurium*) | LT2 (S-strain) | Gift from Dr. H. Nikaido (University of California, Berkeley) | | LPS isolation |
| Strain, Strain Background (*Mus musculus*; Female) | BALB/c (*H-2d*) | Institute of Microbiology and Institute of Physiology of the Czech Academy of Sciences | | |

*Continued on next page*

*Continued*

| Reagent type (species) or resource | Designation | Source or reference | Identifiers | Additional information |
|---|---|---|---|---|
| Strain, Strain Background (*Mus musculus*; Male) | C57BL/6 (*H-2b*) | Institute of Microbiology and Institute of Physiology of the Czech Academy of Sciences | | |
| Strain, Strain Background (*Mus musculus*; Female) | CD1 nude mice (Nu/Nu) | Animal facility of Masaryk University, Czech Republic | | |
| Strain, Strain Background (*Mus musculus,* C57BL/6 J) | Transgenic OVA-specific T cells (OT-I) mice | Institute of Microbiology of the Czech Academy of Sciences | | |
| Strain, Strain Background (*Mus musculus,* C57BL/6 J) | Transgenic OVA-specific T cells (OT-II) mice | Institute of Microbiology of the Czech Academy of Sciences | | |
| Strain, Strain Background (*Mus musculus*; C57BL/6 J) | IFN gamma deficient (*Ifng*$^{-/-}$) | Institute of Microbiology of the Czech Academy of Sciences | | |
| Strain, Strain Background (*Mus musculus*; C57BL/6 J) | MyD88 deficient (*Myd88*$^{-/-}$) | Institute of Molecular Genetics of the Czech Academy of Sciences | | |
| Strain, Strain Background (*Mus musculus*; C57BL/6 J) | Rag1 deficient (*Rag1*$^{-/-}$) | Institute of Molecular Genetics of the Czech Academy of Sciences | | |
| Strain, Strain Background (*Mus musculus,* C57BL/6 J) | B6.SJL (Ly5.1) | Institute of Microbiology of the Czech Academy of Sciences | | |
| Peptide, Recombinant Protein | Recombinant Murine IL-2 | PeproTech | Cat#: 212–12 | Purified from *E. coli* |
| Commercial Assay or Kit | Mouse TNF-alpha DuoSet ELISA | R&D Systems | Cat#: DY410 | |
| Commercial Assay or Kit | Mouse IFN-gamma DuoSet ELISA | R&D Systems | Cat#: DY485 | |
| Commercial Assay or Kit | Mouse IL-1 beta/IL-1F2 DuoSet ELISA | R&D Systems | Cat#: DY401 | |
| Commercial Assay or Kit | Mouse IL-12 p70 DuoSet ELISA | R&D Systems | Cat#: DY419 | |
| Commercial Assay or Kit | Mouse IL-6 DuoSet ELISA | R&D Systems | Cat#: DY406 | |
| Chemical Compound, Drug | Brefeldin A | Sigma-Aldrich | Cat#: B7651 | 150 µg/mouse |
| Software, Algorithm | FlowJo | Tree Star | RRID: SCR_008520 | |
| Software, Algorithm | GraphPad Prism | GraphPad Software | RRID: SCR_002798 | |
| Other | ACK Lysing Buffer | Thermo Fisher Scientific | Cat#: A1049201 | Red blood cell lysis |
| Other | Collagenase D | Sigma-Aldrich (Roche) | Cat#: 11088858001 | 1 mg/mL |

Continued

| Reagent type (species) or resource | Designation | Source or reference | Identifiers | Additional information |
|---|---|---|---|---|
| Other | Foxp3/ Transcription Factor Fixation/Permeabilization Concentrate and Diluent | eBioscience | Cat#: 00-5521-00 | |
| Other | Polyinosinic-polycytidylic acid potassium salt (Poly I:C) | Sigma-Aldrich | Cat#: P9582 | 75 µg/mouse |
| Other | 5-Bromo-2'-deoxyuridine (BrdU) | Sigma-Aldrich (Roche) | Cat#: 10280879001 | 0.5 µg/mouse i.p., 0.8 mg/mL p.o. |
| Other | TRIzol Reagent | Thermo Fisher Scientific (Invitrogen) | Cat#: 15596026 | |
| Other | TURBO DNase | Thermo Fisher Scientific | Cat#: AM2238 | |
| Other | SuperScript IV Reverse Transcriptase | Thermo Fisher Scientific | Cat#: 18090010 | |

## Mice

BALB/c (*H-2d*) and C57BL/6 (*H-2b*) mice were obtained from a breeding colony at the Institute of Microbiology or Institute of Physiology of the Czech Academy of Sciences (Prague, Czech Republic). Athymic CD1 nude mice (Nu/Nu) were acquired from the animal facility of Masaryk University (Brno, Czech Republic). Transgenic OT-I and OT-II mice, *Ifng*[-/-], *Myd88*[-/-] and *Rag1*[-/-] mice and B6.SJL (Ly5.1) mice were bred and kept at mouse facilities at the Institute of Molecular Genetics or the Institute of Microbiology of the Czech Academy of Sciences (Prague, Czech Republic). Mice were used at 9–15 weeks of age.

## Monoclonal antibodies (MAbs)

The following anti-mouse mAbs were used for surface staining during flow cytometry: CD3-eFluor 450, CD8-PerCP-Cy5.5, CD11b-AlexaFluor 700, CD11b-eFluor 450, CD14-FITC, CD25-APC, CD25-eFluor 450, CD45.1-APC, Ly6C-APC-eFluor 780, I-A/E-FITC (eBioscience, San Diego, CA, USA), CD45.1-eFluor 450, CD45R-Horizont V500, CD3-Horizont V500, CD4-Horizont V500, CD4-PerCP, CD8-Horizont V500, Ly6G-AlexaFluor 700, Siglec F-PE (BD Biosciences, San Jose, California, USA) and TLR4-APC (BioLegend, San Diego, Ca, USA). For intracellular staining, the following anti-mouse mAbs were used: TNFα-PE, Foxp3- and IFNγ-PE (eBioscience; San Diego, California, USA). Fc-block (anti-CD16/CD32 mAbs; eBioscience, San Diego, California, USA) was used both in surface and intracellular staining. For ELISA, matched anti-mouse capture mAb and biotinylated detection mAb against TNF-α, IFN-γ, IL-1β, IL-12 and IL-6 were used (R&D System, Minneapolis, Minnesota, USA). Blocking anti-mouse CD25 (PC61.5), CD4 (GK1.5) and anti-IFN-γ (XMG1.2) mAbs, as well as anti-mIL-2 mAbs for preparing IL-2 complexes (S4B6, JES6-1A12), were obtained from BioXcell (Lebanon, New Hampshire, USA).

## IL-2/JES6-1 and IL-2/S4B6 complexes

Complexes were prepared by mixing recombinant mouse IL-2 (Peprotech, Cranbury, New Jersey, USA) with anti-IL-2 mAb S4B6 or JES6-1A12 (both reagents were in PBS) at molar ratio 2:1. After 15 min (min) incubation at room temperature, the complexes were diluted with PBS to the desired concentration before application.

## Isolation of LPS

Bacterial culture (*S. typhymurium* LT2, S-strain; a kind gift from Dr. H. Nikaido, University of California, Berkeley, USA) was first cultivated on agar (Nutrient agar nr.2, HiMedia Laboratories, Mumbai, India) in a petri dish and then transferred to an Erlenmayer flask with liquid media (150 ml, MRS Broth, Oxoid, Hampshire, UK). The bacterial culture was cultivated at 37 °C for up to 24 h (h) in order to reach the exponential growth phase according to measured OD ( ~ 5 × $10^8$/ml bacteria). Next, it was transferred to the solid substrate (12 g Nutrient agar nr.2, 3.5 g agar, 15 g glucose, 300 ml water) in 2 l glass

cultivation bottles by Roux (4 ml inoculum/bottle). Bottles were cultured for 24 h at 37 °C. Bottles were washed with PBS and bacteria were removed from the surface of the substrate with a glass rod. The acquired suspension was filtered through the gauze, and bacteria were killed with 2% phenol (Sigma-Aldrich, St. Louis, Missouri, USA) in ethanol (96%; Lachner, Neratovice, Czech Republic) and deionized water at room temperature. Viability of the bacteria was tested by plating on a new agar petri dish (Nutrient agar nr.2, HiMedia Laboratories, Mumbai, India). Bacteria were sequentially centrifuged at 4600 and 16,000 rpm (room temperature, 20 min). They were washed once with PBS followed by a triple wash with deionized water to remove salts and phenol before they underwent lyophilization in order to get dry matter. Crude LPS was acquired from dry matter by fractionation. This was achieved by the addition of preheated phenol and deionized water at 68 °C (water bath) for 15 min while stirring constantly (20 g of dry matter + 350 ml deionized water and 350 ml 90% phenol). After incubation, the sample was cooled down to 4 °C and sequentially centrifuged at 4600 rpm and 16,000 rpm (4 °C, 20 min). The water phase (top part) was removed and stored whilst the pellet was subjected to a second round of fractionation. Water phases from both separations were pooled, shaken with diethylether (600 ml; Lachner, Neratovice, Czech Republic) and left overnight at 4 °C, followed by repeated dialysis of the water phase for 3–5 days (d) in a huge excess of deionized water. The whole process of crude LPS isolation was finalized by lyophilization. To achieve this, 0.76 g of crude LPS was dissolved in 50 ml of deionized water and cooled down to 4 °C. Next, 150 ml of ice-cold methanol (Lachner, Neratovice, Czech Republic) and 1.5 ml 20% $MgCl_2$ (Sigma-Aldrich, St. Louis, Missouri, USA) were added and it was left stirring overnight. The following day, the solution was centrifuged at 5000 rpm (4 °C, 20 min) and the pellet was resuspended again in 50 ml of deionized water and cooled down to 4 °C. After this, 150 ml of ice-cold methanol was added and it was left stirring overnight. On the following day, the same procedure was repeated and the pellet was resuspended in 20 ml of deionized water. Pure LPS was finally lyophilized.

## LPS challenge in vivo

Mice were treated i.p. with IL-2/JES6-1 or IL-2/S4B6 complexes daily for 3 d (1.5 µg IL-2/mouse in 250 µl) as shown in *Figure 1A*, unless stated otherwise. After 48 h, mice were i.p. injected with LPS. Control mice were injected with the same volume of sterile PBS (250 µl). The dose of injected LPS is shown in % of maximum non-lethal dose (MNLD; 100% MNLD ~ 200, 35 and 35 µg LPS in C57BL/6, BALB/c and Nu/Nu mice, respectively), that is the highest possible dose which causes significant toxic effect, but no mortality. Body temperature (measured by stylus in the throat) and survival of mice were recorded.

## Preparation of single-cell suspension for flow cytometry

For spleen cells, animals were sacrificed by cervical dislocation and spleens were harvested and homogenized by GentleMACS Dissociator (Miltenyi Biotec, Bergisch Gladbach, North Rhine-Westphalia, Germany). After red blood cell lysis (ACK lysing buffer, Gibco, Gaithersburg, Maryland, USA), cells were strained twice to remove clumps (70 µm BD Falcon strainer, Corning, New York, New York, USA, followed by 30 µm CellTrics strainer, Sysmex, Norderstedt, Germany) and resuspended in FACS buffer (PBS, 2% FCS, 2 mmol EDTA). For peripheral blood cells, mice were exsanguinated via carotid excision into the heparinized Eppendorf tubes. Whole blood was lysed twice with ACK Lysing buffer, strained once (30 µm, Sysmex Norderstedt, Germany) and resuspended in FACS buffer. Lungs and livers were injected with Colagenase D solution (1 mg/ml, Roche, Basel, Switzerland), incubated for 30 min at 37 °C, homogenized by GentleMACS Dissociator and further processed like the spleen cells.

## Staining for surface markers

Cells were blocked by 10% mouse serum and/or Fc block for 30 min on ice and stained with fluorochrome-labelled mAbs recognizing selected surface markers for 30 min on ice in the dark. After each step, cells were washed twice in FACS buffer and fixed with Foxp3 Fixation/Permeabilization buffer (eBiosciences, San Diego, California, USA) for 30 min on ice in the dark before analysis. Flow cytometric analysis was performed on LSRII (BD Biosciences, San Jose, California, USA) and data were analysed using FlowJo X software (Tree Star, Inc, Ashland, Oregon, USA).

## Staining for intracellular markers

After staining of surface antigens, fluorochrome-labelled mAbs recognizing intracellular markers were added and cells were incubated for 30 min on ice in the dark. Washing was performed twice with Fixation/Permeabilization buffer (eBiosciences, San Diego, California, USA). Cells were resuspended in FACS buffer before analysis. Analysis was performed as described above.

## RT-qPCR analysis

RNA from splenocytes was isolated using TRIzol reagent (Invitrogen, USA) regarding to the manufacturer's protocol. One µg RNA treated with DNAse I (Turbo DNAse; Thermo Fisher Scientific, USA) was reverse-transcribed using the Oligo(dT)12–18 primer and Superscript IV Reverse Transcriptase (Life Technologies, Carlsbad, CA, USA). To select suitable housekeeping genes, the gene expression stability was assessed using RefFinder (https://www.heartcure.com.au/reffinder/). Evaluation of amplification efficiency was performed by dilution series of cDNA. qPCR (CFX96 TouchTM, Bio-Rad) was performed to determine the changes in the mRNA levels of TLR4. The cycling parameters were as follows: 4 min at 94 °C, 35 cycles of 10 s at 94 °C, 25 s at 58 °C, and a final extension for 7 min at 72 °C. Gene expression changes were calculated according to the $2-\Delta\Delta CT$ (Livak) method. Two reference genes (*Casc3, H6pd*) were selected as the most stable internal controls for the normalization of the *Tlr4* gene expression. The fold change in the mRNA level was related to the change in the settled controls. All parameters were determined in duplicates. Primers used:

| Name | Sequence (5´- 3´) | Target |
|---|---|---|
| *Casc3*for | TTCGAGGTGTGCCTAACCA | |
| *Casc3*rev | GCTTAGCTCGACCACTCTGG | *Casc3* |
| *H6pd*for | GGATTGTGTTTAAGAATCGGG | |
| *H6pd*rev | AGTAGGCGTCTTGCTC | *H6pd* |
| *Tlr4*for | GATCATGGCACTGTTCTTCTC | |
| *Tlr4*rev | CACACCTGGATAAATCCAGC | *Tlr4* |

*Casc3*: cancer susceptibility candidate gene 3; *H6pd*: Hexose-6-phosphate dehydrogenase; *Tlr4*: Toll-like receptor 4.

## Adoptive transfer of OT-I CD8+ T cells, OT-II CD4+ T cells, CD4+CD25+ T cells and CD3+ T cells

MACS-separated (negative selection, AutoMACS, Miltenyi Biotec, Bergisch Gladbach, North Rhine-Westphalia, Germany) OT-I CD8+ T and OT-II CD4+ cells (both Ly5.1+) were injected i.v. into C57BL/6 (Ly5.2+) mice via tail vein. C57BL/6 mice were injected i.p. with PBS, OT-I plus OT-II peptides (10 and 50 µg/mouse, respectively; MBL International, Woburn, Massachusetts, USA; or Genscript, Piscataway, New Jersey, USA, respectively) plus polyI:C (75 µg/mouse; Sigma-Aldrich, St. Louis, Missouri, USA) or with the latter plus αIFN-γ mAb (250 µg/mouse; XMG1.2; BioXcell, Lebanon, New Hampshire, USA). Mice were either sacrificed 3 days post priming and spleen cells were analyzed by flow cytometry for expansion of adoptively transferred cells or challenged with LPS. CD4+CD25+ T cells were purified by MACS (negative selection for CD4+ T cells and positive selection for CD25+ cells) from C56BL/6 mice treated with IL-2/JES6 as shown in *Figure 1A* (2 days after the last dose of IL-2/JES6). CD4+CD25+ T cells were injected i.v. into Nu/Nu mice via tail vein. Next, Nu/Nu mice were treated with IL-2/JES6 and challenged with LPS as in *Figure 1A*. CD3+ T cells were purified by MACS (negative selection) from MyD88-/- mice treated with IL-2/JES6 as shown in *Figure 1A* (2 days after the last dose of IL-2/JES6). CD3+ T cells were injected i.v. into Rag1-/- mice via tail vein. Next, Rag1-/- mice were treated with IL-2/JES6 and challenged with LPS as in *Figure 1A*.

## Immunization with ovalbumin

C57BL/6 mice were i.p. injected with ovalbumin (OVA, 0.5 mg/mouse, Warthington, New Jersey, USA) plus polyI:C (75 µg/mouse) on day 0. Mice were i.p. challenged with LPS on day 4.

### BrdU incorporation assay in vivo

C57BL/6 mice were i.p. injected with titrated doses of IL-2/JES6 or with PBS. After 4 h, mice were injected i.p. with 0.5 µg of BrdU (Sigma-Aldrich, St. Louis, Missouri, USA) in 50 µl PBS. At the same time, mice were given 0.8 mg/ml BrdU in their drinking water. BrdU solution was prepared in sterile water, protected from light exposure and changed daily. Mice were sacrificed 48 h later and spleens were harvested. Single cell suspensions were prepared as described above. BrdU staining was performed in similar fashion to staining of intracellular antigens with two additional steps following fixation with Fixation/Permeabilization buffer (eBiosciences, San Diego, California, USA). First, cells were treated with BD Cytofix/Cytoperm Plus buffer (BD Biosciences, San Jose, California, USA) for 10 min on ice. Second, cells were treated with DNAse (BD Biosciences, San Jose, California, USA) for 1 h at 37 °C. Fluorochrome-labelled mAbs were added together with anti-BrdU mAb (BD Biosciences, San Jose, California, USA) and cells were incubated for 30 min on ice in the dark. Washing, resuspending of cells and subsequent analysis was performed as described above.

### Detection of IFN-γ production in vivo

C57BL/6 mice were i.p. injected with IL-2/JES6-1, IL-2/S4B6 or PBS (Control) as in *Figure 1A*. Each mouse was i.p. injected with 150 µg of brefeldin A (Sigma-Aldrich, St. Louis, Missouri, USA) 2 h after the last dose. Mice were sacrificed 12 h after injection of brefeldin A. Spleens, livers, and lungs were harvested for subsequent flow cytometry analysis performed as described above.

### Detection of cytokines in serum

C57BL/6 and BALB/c mice were treated with IL-2/JES6 and challenged with LPS (10% of MNLD) as shown in *Figure 1A*. Control mice were treated with sterile PBS and challenged with LPS (100% of MNLD). Mice were euthanized by exsanguination via carotid excision at 2, 3, 4, 6, and 8 h after the LPS challenge and their sera were collected. Levels of TNF-α, IL-1β, IL-12, and IL-6 (R&D System, Minneapolis, Minnesota, USA) following manufacturer's protocol.

### Determination of vascular leak syndrome (VLS) in the lungs

C57BL/6 mice were treated with IL-2/JES6, IL-2/S4B6 or PBS (Control) as shown in the *Figure 1A*. Mice were euthanized 1 day after the last dose and their lungs were harvested. Pulmonary wet weight was determined by weighting lungs before and after lyophilization overnight at 58 °C under vacuum.

### Statistical analysis

All experiments were done at least twice with similar results; n = 2–16 technical replicates. Statistical analysis was performed using GraphPad Prism (GraphPad Software, San Diego, California, USA). The difference between groups was analysed by unpaired two-tailed Student's $t$-test. The confidence level was 95%. Differences with *,°,+ $p \leq 0.05$; **, °° $p \leq 0.01$; ***, °°° $p \leq 0.001$ were considered as statistically significant.

## Acknowledgements

We thank to Helena Misurcova and Pavlina Jungrova for excellent technical help and to Dr. H Nikaido (University of California, Berkeley, USA) for providing *S. typhymurium* LT2, S-strain. We thank to Dr. Dominik Filipp (Institute of Molecular Genetics, Czech Academy of Sciences, Prague, Czech Republic) for kind providing Rag1[-/-] and MyD88[-/-] mice. We thank to the Czech Centre for Phenogenomics (Czech Academy of Sciences, Vestec, Czech Republic) for the help with breeding the MyD88[-/-] mice. We also thank to Tomas Etrych and his laboratory from Institute of Macromolecular Chemistry (Czech Academy of Sciences, Prague, Czech Republic) for providing HPMA copolymer-bound IL-2.

## Additional information

### Funding

| Funder | Grant reference number | Author |
| --- | --- | --- |
| Czech Science Foundation | 13-12885S | Jakub Tomala<br>Petra Weberova<br>Barbora Tomalova<br>Ladislav Sivak<br>Marek Kovar |
| Institutional Research Concept RVO | 61388971 | Jakub Tomala<br>Petra Weberova<br>Barbora Tomalova<br>Zuzana Jiraskova Zakostelska<br>Ladislav Sivak<br>Jirina Kovarova<br>Marek Kovar |
| Czech Science Foundation | 18-12973S | Jakub Tomala<br>Petra Weberova<br>Barbora Tomalova<br>Ladislav Sivak<br>Marek Kovar |
| Ministry of Health of the Czech Republic | NV19-03-00179 | Zuzana Jiraskova Zakostelska |

The funders had no role in study design, data collection and interpretation, or the decision to submit the work for publication.

### Author contributions

Jakub Tomala, Data curation, Investigation, Methodology, Validation, Writing - original draft; Petra Weberova, Investigation, Methodology, Validation; Barbora Tomalova, Data curation, Formal analysis, Investigation, Methodology, Validation, Visualization, Writing - original draft, Writing - review and editing; Zuzana Jiraskova Zakostelska, Data curation, Investigation, Methodology; Ladislav Sivak, Data curation, Formal analysis, Investigation, Visualization; Jirina Kovarova, Investigation, Methodology; Marek Kovar, Conceptualization, Data curation, Formal analysis, Funding acquisition, Investigation, Project administration, Supervision, Visualization, Writing - original draft, Writing - review and editing

### Author ORCIDs

Jakub Tomala http://orcid.org/0000-0002-6315-2832
Ladislav Sivak http://orcid.org/0000-0003-2623-8458
Marek Kovar http://orcid.org/0000-0002-6602-1678

### Ethics

All experiments were approved by the Animal Welfare Committee at the Institute of Microbiology of the Czech Academy of Sciences (Prague, Czech Republic) and performed in strict accordance with the recommendations of local and European guidelines. Every effort was made to minimize animal suffering. This work was approved by protocol number 112/2012. Accreditation number of the animal facility where the research was conducted: 51205/2018-MZE-17214.

### Decision letter and Author response

Decision letter https://doi.org/10.7554/eLife.62432.sa1
Author response https://doi.org/10.7554/eLife.62432.sa2

## Additional files

### Supplementary files

• Transparent reporting form

### Data availability

All data generated or analysed during this study are included in the manuscript and supporting files. Source data files have been provided for Figures 1-6 and all figure supplements (12 in total).

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
