## [Editor Report]

Tomala et al., describes the ability of IL-2/JES6-1 mAb complexes to increase mouse sensitivity to LPS challenge. The authors present data to suggest this is due to IFN-γ production by CD25^+^Foxp3^-^ T cells. The manuscript has identified an interesting phenomenon as a result of IL-2/JES6-1 complex administration. These data may provide novel avenues for future therapeutic intervention in autoimmune disease.

---

## [Decision Letter]

**Decision letter after peer review:**

Thank you for submitting your article "IL-2/JES6-1 mAb complexes dramatically increase sensitivity to LPS through IFN-γ production by CD25+Foxp3- T cells" for consideration by *eLife*. Your article has been reviewed by 2 peer reviewers, and the evaluation has been overseen by a Reviewing Editor and Jos van der Meer as the Senior Editor. The following individuals involved in the review of your submission have agreed to reveal their identity: Thomas S Griffith (Reviewer #1); Angel Porgador (Reviewer #3).

The reviewers have discussed the reviews with one another and the Reviewing Editor has drafted this decision to help you prepare a revised submission.

Tomala et al., describe the capability of IL-2/JES6-1 mAb complexes to increase mouse sensitivity to LPS challenge. These data suggest this is due to IFN-g production by CD25+Foxp3- T cells. The manuscript has identified an interesting phenomenon as a result of IL-2/JES6-1 complex administration, but the attempt to mechanistically explain why LPS sensitization occurs has fallen short of a complete and convincing story. The authors will need to address a number of concerns to progress, please see below.

Essential revisions:

1. In the introduction an explanation as to how IL2/JES6 recruits the IL-2R is discused. This explanation is obviously preparing the reader to assume that IL2/JES6 would not sensitize LPS reaction. However, the paper demonstrates, this is not the case. The authors also show the necessity of CD25 and compared the response to native IL-2, attributing the difference due to the short life-span of IL-2 (which is speculative) and needs to be addressed. Assuming the explanation in the introduction is true, it is hard to understand how a complex which primarily binds CD25, and only recruits IL-2Rb and IL-2Rg rarely, is so efficient for the induction of CD25+FoxP3- cells, which are more dependent on IL-2Rb and IL-2Rg chain activation.

2. The transient nature of the IL-2/JES6-1 complex-induced sensitization to LPS (Figure 1C) is an interesting observation that was not followed up on and why does this occur?

3. The examination of TLR4 expression on different immune cell population following IL-2/JES6-1 complex administration would be telling, for example, Is the increased sensitivity to LPS due to increased number of cells (such as myeloid cells) responsive to LPS or increased TLR4 expression or both?

4. A more detailed examination of the myeloid cells in Figure 3 is required, to define if these cells are monocytes, macrophages, neutrophils, or another cell type?

5. Figure 4 F-G demonstrates the effect of IL-2/JES6-1 complex on the T cell compartment, however, this only assesses the proliferative response of these cells. The focus on the CD25+Foxp3- CD4 and CD8 T cells in Figure 5 is not well justified. What happens to the number of CD25+Foxp3+ CD4 T cells?

6. In Figure 6 IL-2/JES6-1 complexes were overlooked in the mice that received the activated OT-I and OT-II T cells? The authors transferred 2 and 4 million OT-I and OT-II T cells, respectively, prior to in vivo peptide stimulation. This number is exceeding higher than the number of endogenous OVA-specific CD8 and CD4 T cells. What happens to naïve mice immunized with OVA that do not receive any OT-I or OT-II cells? There are a lot of steps bypassed between IL-2/JES6-1 administration and IFNγ production by T cells after LPS administration. Are the T cells being directly stimulated by the LPS? The authors should transfer TLR4-/- T cells to see what happens.

7. IL-2 treatment had a toxic side effect, especially as an inducer of vascular leak syndrome (VLS). Vascular leakage is also attributed to LPS and IFNγ inflammatory response. Did the authors consider showing a manifestation of VLS in the lungs of mice treated with IL2/JES6 compared to other treatments? Showing, or describing previous work, the outcome of this well-documented phenomena following the injection of IL2/JES6 will give a clinical significance to the findings presented.

---

## [Author Response]

Essential Revisions:1. In the introduction an explanation as to how IL2/JES6 recruits the IL-2R is discused. This explanation is obviously preparing the reader to assume that IL2/JES6 would not sensitize LPS reaction. However, the paper demonstrates, this is not the case. The authors also show the necessity of CD25 and compared the response to native IL-2, attributing the difference due to the short life-span of IL-2 (which is speculative) and needs to be addressed. Assuming the explanation in the introduction is true, it is hard to understand how a complex which primarily binds CD25, and only recruits IL-2Rb and IL-2Rg rarely, is so efficient for the induction of CD25+FoxP3- cells, which are more dependent on IL-2Rb and IL-2Rg chain activation.

We thank the reviewer for interesting question. We believe that the explanation why sensitization to LPS by IL-2 alone (free/native IL-2) is much weaker in comparison to IL-2/JES6, i.e. considerable extension of half-life *and* selectivity for CD25^+^ cells (two separate features), is true and based on data shown within the manuscript. The fact that IL-2 has very short half-life (5 to 7 min in humans) is mentioned elsewhere and we decided to introduce a new reference into revised manuscript to support this statement (page 3, line 52). We employed IL-2 bound to synthetic water-soluble and fully biocompatible polymer carrier based on poly(*N*-(2-propyl)methacrylamide) (IL-2-pHPMA) in our study to dissect half-life extension feature and CD25^+^ cell selectivity feature of IL-2/JES6. IL-2-pHPMA possesses much higher half-life in comparison to IL-2 and very similar to that of IL-2/JES6 (3-4 h; mentioned on page 7, line 140 including the reference to our publication where half-life of IL-2-pHPMA was determined, line 154) but lacks CD25^+^ cell selectivity. IL-2-pHPMA showed more profound sensitization to LPS in comparison to IL-2 (Figure 2A and B) even when IL-2 was administered at much higher dosage (35 versus 1.5 µg/dose). This shows that prolongation of the half-life plays an important role. Nevertheless, sensitization to LPS via IL-2-pHPMA was not as robust as with IL-2/JES6 showing that selectivity for CD25^+^ cells is also important. This was proved by the use of anti-CD25 mAb, which totally abolished sensitization to LPS via IL-2/JES6 (Figure 2E) and by comparison of sensitization to LPS via IL-2/JES6 and IL-2/S4B6 (CD25 independent IL-2 complex, i.e. the same half-life but different selectivity; Figure 1 B and D). Thus, we believe that this explanation supports our conclusion that both the extension of the half-life as well as selective stimulatory activity for CD25^+^ cells are key features of IL-2/JES6 governing the sensitization to LPS.

IL-2/JES6 requires CD25 to be expressed on the cell surface in order to enable cells to utilize IL-2 from IL-2/JES6 complex. However, it is not completely true that IL-2/JES6 recruits IL-2Rβ and IL-2Rγ “rarely”. CD25 enables the dissociation of IL-2/JES6 complex via “peeling off” the IL-2 molecule from the JES6 antibody and binding the IL-2 (Spangler et al., *Immunity*, 42: 815-825, 2015; cited in the manuscript, page 4, line 75). Next, CD25-bound IL-2 engages IL-2Rβγ dimer to induce signaling. CD25 contains only short cytoplasmic tail with no motive to bind any signaling molecules (e.g. JAK kinases). Thus, IL-2/JES6 complex always utilizes IL-2Rβγ dimer to provide IL-2 signal to CD25^+^ IL-2 responsive cells, however, after dissociation of the cytokine/antibody complex and transfer of the IL-2 to CD25 molecule. IL-2/JES6 complex is “invisible” for CD25 negative cells expressing only IL-2Rβ and IL-2Rγ since JES6 mAb binds to epitope in IL-2 molecule which almost completely overlaps with binding sites for IL-2Rβ and IL-2Rγ. This explains why IL-2/JES6 is so efficient in the stimulation of CD25^+^Foxp3^-^ T cell proliferation: only CD25^+^ cells are able to compete for IL-2 in the form of IL-2/JES6. Moreover, CD25^+^Foxp3^-^ T cells are very keen to proliferate upon IL-2 signaling as shown by BrdU incorporation (Figure 4F and G). We do not presume that IL-2/JES6 induces CD25^+^Foxp3^-^ T cells de novo (from CD25 negative T cells) but it rather potently expands already existing very tiny population of these cells (~ 0.1 %, Figure 4F and G). To make it clear, we decided to rephrase the sentence:

“Nevertheless, we found that administration of IL-2/JES6 gave rise to CD25^+^Foxp3^-^ T cells in both CD4^+^ and CD8^+^ subsets …….”

to the following one:

“Nevertheless, we found that administration of IL-2/JES6 potently expanded CD25^+^Foxp3^-^ T cells in both CD4^+^ and CD8^+^ subsets …….” (page 10, lines 228-231).

Further, we rephrased the sentence:

“We focused on CD25^+^Foxp3^-^CD4^+^ and CD8^+^ T cells induced by IL-2/JES6 treatment”

as follows:

“We focused on CD25^+^Foxp3^-^CD4^+^ and CD8^+^ T cells robustly expanded in the spleen of mice treated with IL-2/JES6.” (page 11, line 236-237).

We hope that this makes the issue more comprehensible for the readers and we are very sorry for a confusing statement in the previous version of our manuscript.

2. The transient nature of the IL-2/JES6-1 complex-induced sensitization to LPS (Figure 1C) is an interesting observation that was not followed up on and why does this occur?

The study on kinetics of sensitization to LPS via IL-2/JES6 was performed to determine whether the standard time schedule selected for LPS administration (Figure 1A) in ongoing experiments is appropriate in terms of maximal sensitivity to LPS. The experiment shown in Figure 1C confirmed that administration of LPS 2 days after the last dose IL-2/JES6 is still at the plateau of the maximal sensitivity to LPS. Thus, this kinetics experiment provided us a clear answer to our question and we did not see any need to continue in this kind of experiments (however, we completely agree with the reviewer that the observation is interesting). We think that the transient nature of the IL-2/JES6-induced sensitization to LPS could be explained by short-term duration of IFN-γ production (predominantly by CD25^+^Foxp3^-^ CD8^+^ and CD4^+^ T cells) upon IL-2/JES6 administration and sensitization of myeloid cells (increased in counts in IL-2/JES6 treated mice) to LPS by such produced IFN-γ. Thus, the production of IFN-γ probably rapidly diminishes after the last dose of IL-2/JES6 leading to the loss of increased sensitivity to LPS after about 4 days as shown in Figure 1C.

3. The examination of TLR4 expression on different immune cell population following IL-2/JES6-1 complex administration would be telling, for example, Is the increased sensitivity to LPS due to increased number of cells (such as myeloid cells) responsive to LPS or increased TLR4 expression or both?

We definitely agree with the reviewer that it would be useful to investigate TLR4 expression on LPS responsive cells upon IL-2/JES6 treatment. Thus, we treated B6 mice with IL-2/JES6 or IL-2/S4B6 as shown in the Figure 1A and analyzed first the *Tlr4* expression in splenocytes of these mice by quantitative RT-PCR. We found that neither treatment affected the expression of *Tlr4* on mRNA level. Next, we analyzed TLR4 expression in various subsets of myeloid cells from the spleen of B6 mice treated as described above via flow cytometry. Surprisingly, we found decent but still statistically significant decrease of TLR4 levels in CD11b^+^Ly6G^+^ (granulocytes) and CD11b^+^Ly6G^-^Ly6C^low^ cells upon IL-2/JES6 treatment in comparison to controls. No statistically significant difference in TLR4 level upon IL-2/JES6 treatment in comparison to controls was found in CD11b^+^Ly6G^-^Ly6C^high^ cells (monocytes/macrophages). Thus, we conclude that increased sensitivity of B6 mice treated with IL-2/JES6 to LPS is not due to the increased TLR4 expression but rather due to the increased counts of cells responsive to LPS as shown in the manuscript. The results described above are now included in the revised version of the manuscript (Figure 3 —figure supplement 4 and 5). We also added the following text (page 9, lines 185-194):

“We asked a question whether this increased responsiveness of myeloid cells to LPS was due to the increased expression of TLR4. Thus, we analyzed *Tlr4* expression in spleen cells via quantitative RT-PCR and in various myeloid cell subsets via flow cytometry. Treatment with IL-2/JES6 did not affect the *Tlr4* expression in splenocytes on mRNA level (Figure 3 —figure supplement 4). No statistically significant difference in TLR4 level upon IL-2/JES6 treatment in comparison to control was found in CD11b^+^Ly6G^-^Ly6C^high^ cells. Surprisingly, IL-2/JES6 treatment decently but statistically significantly decreased TLR4 levels in CD11b^+^Ly6G^+^ cells and CD11b^+^Ly6G^-^Ly6C^low^cells (Figure 3 —figure supplement 5). IL-2/JES6 thus did not increased the sensitivity to LPS via increased expression of TLR4.”

4. A more detailed examination of the myeloid cells in Figure 3 is required, to define if these cells are monocytes, macrophages, neutrophils, or another cell type?

We thank the reviewer for this suggestion and we completely agree that a more detailed examination of the myeloid cells expanded in response to IL-2/JES6 administration would improve the manuscript. Thus, we performed an experiment where we treated B6 mice with IL-2/JES6 or IL-2/S4B6 as shown in the Figure 1A and analyzed the myeloid cells from the spleen by flow cytometry 2 d after the last dose of IL-2 complexes (at time when LPS is injected in other experiments). We also confirmed proliferation of myeloid cells upon various single doses of IL-2/JES6 via BrdU incorporation. These results are now included in revised version of the manuscript (Figure 3 —figure supplements 1, 2 and 3). We also added the following text (page 8, lines 173-179):

“We also found that treatment with IL-2/JES6 expands myeloid cells in general. IL-2/JES6 expanded significantly granulocytes, eosinophils and DCs and elevated, though not significantly, relative counts of monocytes and macrophages (Figure 3 —figure supplement 1). Of note, treatment with IL-2/JES6 increased MHC II expression on monocytes and macrophages (Figure 3 —figure supplement 2). The proliferation of myeloid cells driven by IL-2/JES6 was further confirmed by BrdU incorporation (Figure 3 —figure supplement 3).”

5. Figure 4 F-G demonstrates the effect of IL-2/JES6-1 complex on the T cell compartment, however, this only assesses the proliferative response of these cells. The focus on the CD25+Foxp3- CD4 and CD8 T cells in Figure 5 is not well justified. What happens to the number of CD25+Foxp3+ CD4 T cells?

The primary purpose we show Figure 4F and G is to demonstrate that treatment with IL-2/JES6 dramatically expands CD25^+^Foxp3^-^CD4^+^ and CD8^+^ T cells (originally very tiny population representing ~ 0.1 % of these T cell subsets, respectively) through potent stimulation of their proliferation as shown by BrdU incorporation assay. These cells proliferate even more vigorously in response to IL-2/JES6 than T_reg_ cells as shown in Figure 4F and G (mentioned in the manuscript, page 10, lines 228-231). Such finding was made for the first time in the spleen and later on the robust expansion of CD25^+^Foxp3^-^CD4^+^ and CD8^+^ T cells after IL-2/JES6 treatment was found also in other (non-lymphoid) organs, particularly in the liver and lungs (shown in the Figure 5A and B). We agree with the reviewer that focus on CD25^+^Foxp3^-^CD4^+^ and CD8^+^ T cells was not well justified in the previous version of our manuscript. Thus, we rewrote the beginning of Results section “IL-2/JES6 drive expansion of CD25^+^Foxp3^-^ T cells producing IFN-γ in lymphoid as well as non-lymphoid tissues” as follows (page 11, lines 236-240):

“We focused on CD25^+^Foxp3^-^CD4^+^ and CD8^+^ T cells robustly expanded in the spleen of mice treated with IL-2/JES6. Since these cells resemble by their phenotype activated T cells, we presumed that these cells could be a key producers of IFN-γ in IL-2/JES6 treated mice causing the increased sensitivity to LPS. Thus, we decided to investigate whether these cells were expanded also in other organs except of spleen and whether they produced IFN-γ.”

The reviewer also asked what happens to the number of CD25^+^Foxp3^+^CD4^+^ T cells (i.e. T_reg_ cells)? T_reg_ cells are also significantly expanded by IL-2/JES6 as shown in Figure 5A, left part of the graph (though the percentage of CD25^+^Foxp3^+^ in CD4^+^ T cell population is not shown) via stimulation of their proliferation (Figure 4F, BrdU incorporation assay part of the panel). The treatment with IL-2/JES6 (using the schedule and dosage shown in Figure 1A) usually expands T_reg_ cells in the spleen to about 25-45 % out of total CD4^+^ T cells. The corresponding number in the liver and spleen is about 10-20 %.

Surprisingly, T_reg_ cells were found also to produce IFN-γ in mice treated with IL-2/JES6, particularly in the liver (Figure 5C and D, middle part of the graphs). This is mentioned in the Results (page 11, lines 251-253) and also in the Discussion (page 14, lines 320-322) where we hypothesize why T_reg_ cells upon IL-2/JES6 administration produce IFN-γ, which is rather unusual and surprising feature of these cells. However, the fraction of IFN-γ producing T_reg_ cells was significantly smaller in comparison to that found in CD25^+^CD8^+^ and CD25^+^Foxp3^-^CD4^+^ T cells. Altogether, we conclude that T_reg_ cells upon IL-2/JES6 administration participate to IFN-γ production in the organism and thus to LPS hypersensitivity, however, the key producers are CD25^+^Foxp3^-^CD4^+^ and CD8^+^ T cells.

6. In Figure 6 IL-2/JES6-1 complexes were overlooked in the mice that received the activated OT-I and OT-II T cells? The authors transferred 2 and 4 million OT-I and OT-II T cells, respectively, prior to in vivo peptide stimulation. This number is exceeding higher than the number of endogenous OVA-specific CD8 and CD4 T cells. What happens to naïve mice immunized with OVA that do not receive any OT-I or OT-II cells? There are a lot of steps bypassed between IL-2/JES6-1 administration and IFNγ production by T cells after LPS administration. Are the T cells being directly stimulated by the LPS? The authors should transfer TLR4-/- T cells to see what happens.

IL-2/JES6 were not overlooked (or missed) in the Figure 6. The data in Figure 6 A-C should demonstrate that potentially any T cells producing IFN-γ, if present in the body at sufficiently high numbers, are capable of sensitization to LPS. We wanted to mimic the situation there is a high number of IFN-γ producing T cells but without the use of IL-2/JES6. Thus, we first show clonal expansion of OT-I CD8^+^ and OT-II CD4^+^ T cells after adoptive transfer into congeneic Ly5.1 mice and treatment of mice with respective cognate peptides plus polyI:C. These cells formed approximately 8 and 4 % of total CD8^+^ and CD4^+^ T cells after their expansion, respectively, and their clonal expansion was not affected by injection of anti-IFN-γ mAb (Figure 6 A and B). These cells should moreover produce IFN-γ due to the TCR stimulation in the presence of adjuvant. Next, we demonstrate that mice with adoptively transferred OT-I CD8^+^ and OT-II CD4^+^ T cells and stimulated with cognate peptides plus polyI:C, but not unstimulated ones, are sensitized to LPS (Figure 6C). Importantly, injection of anti-IFN-γ mAb abrogated the sensitization to LPS. It is true that the number of transferred OT-I CD8^+^ and OT-II CD4^+^ T cells enormously exceeds the number of endogenous OVA-specific T cells (however, it was done intentionally). The reviewer asks what happens to naïve mice immunize with OVA that do not receive any OT-I CD8^+^ and OT-II CD4^+^ T cells? To answer this question, we performed an experiment where we immunized B6 mice with OVA (0.5 mg/mice i.p.) plus poly I:C (75 µg/mice i.p.) and challenged with LPS 4 d after immunization. Positive controls were treated with IL-2/JES6 as in Figure 1A and Control mice were injected with PBS. As expected, mice immunized with OVA plus poly I:C showed no sensitization to LPS. This result is included in the revised version of the manuscript (Figure 6 —figure supplement 1). We also included following sentence into revised version of the manuscript (page 12, lines 271-273):

“C56BL/6 mice without adoptively transferred CD8^+^ OT-I and CD4^+^ OT-II T cells and immunized with OVA plus polyI:C showed no increased sensitivity to LPS (Figure 6 —figure supplement 1).”

The other question of the reviewer was whether T cells could be directly stimulated by LPS and that we should transfer *Tlr4*^-/-^ T cells to see what happens. To address this issue, we did an experiment where we employed *Myd88*^-/-^ mice. We used these mice since we had *Tlr4*^-/-^ mice available on BALB/c background but not on C57BL/6 background. Since the experiment required the transfer of T cells, which would be capable of producing substantial levels of IFN-γ, we decided not to use *Tlr4*^-/-^ mice on BALB/c background since BALB/c mice are known to be a poor IFN-γ responders (contrary to C57BL/6 mice which are a high IFN-γ responders). This is reflected also in sensitization to LPS via IL-2/JES6 as BALB/c mice are sensitized much weaker in comparison to B6 mice (Figure 4B versus Figure 1B). *Myd88*^-/-^ mice have severely impaired TLR4 downstream signaling; only TRIF signaling pathway leading to IRF3 activation is functional whereas IRAK and *TAK1* signaling pathways essential for activation of NF-κB and AP-1 are abrogated. It has been demonstrated that *Myd88*^-/-^ mice are unresponsive to LPS (Kawai et al., Immunity, 11: 115-122, 1999) and thereby we considered them as convenient model for our experiment. Thus, we treated *Myd88*^-/-^ mice on C57BL/6 background for 3 consecutive days with IL-2/JES6 in order to expand CD25^+^Foxp3^-^ T cells (and similarly to experiment using nude mice shown in Figure 4E). Mice were euthanized 1 d after the last dose and T cells were purified from their spleens and lymph nodes by MACS (using negative selection). T cells were adoptively transferred (5.6 x 10^6^ per mice) into *Rag1*^-/-^ mice and the mice were either left untreated or were treated with IL-2/JES6 as in Figure 1A. *Rag1*^-/-^ mice without adoptively transferred T cells either treated with IL-2/JES6 or untreated (Control) were also involved. All experimental groups were challenged with LPS. We found that mice with adoptively transferred T cells and treated with IL-2/JES6 (but no any other groups) showed profound sensitivity to LPS demonstrating that even T cells unable to respond to LPS sensitizes the mice when stimulated with IL-2/JES6. This finding is included in revised version of the manuscript (Figure 4 —figure supplement 1). We also included following sentence into revised version of the manuscript (page 10, lines 221-224):

“TLR4 expression on T cells seems to be irrelevant for their ability to sensitize the mice to LPS since *Rag1*^-/-^ mice with adoptively transferred T cells from *Myd88*^-/-^ mice, i.e. with severely impaired TLR4 downstream signaling, showed profound LPS sensitivity upon treatment with IL-2/JES6 (Figure 4 —figure supplement 1).”

7. IL-2 treatment had a toxic side effect, especially as an inducer of vascular leak syndrome (VLS). Vascular leakage is also attributed to LPS and IFNγ inflammatory response. Did the authors consider showing a manifestation of VLS in the lungs of mice treated with IL2/JES6 compared to other treatments? Showing, or describing previous work, the outcome of this well-documented phenomena following the injection of IL2/JES6 will give a clinical significance to the findings presented.

IL-2 treatment causes vascular leak syndrome (VLS) accompanied by lung oedema, indeed. Lung oedema was described to be a serious side toxicity in IL-2-treated patients and also in mice treated with IL-2 or IL-2/anti-IL-2 mAb complexes. We completely agree with the reviewer that showing lung oedema in mice treated with IL-2/JES6 in comparison to IL-2/S4B6 treated mice (i.e. comparison of IL-2 complex inducing high LPS sensitivity versus IL-2 complex inducing only decent LPS sensitivity) would improve the manuscript and provide clinical significance to our findings. Thus, we performed an experiment where we injected the B6 mice with either IL-2/JES6 or IL-2/S4B6 using schedule and dosage shown in Figure 1A. Mice were euthanized 1 d after the last dose of IL-2 complexes and wet weight of their lungs was determined. The result recapitulates the previously described finding (Krieg et al., PNAS, 109: 11906-1191, 2012) that both IL-2 complexes induce lung oedema but IL-2/JES6 do it significantly more vigorously. This data is now included in the revised version of the manuscript (Figure 1 —figure supplement 2) and mentioned in the Discussion (page 16, lines 368-371):

“Further, treatment with IL-2/JES6 *per se* induces more severe lung oedema in comparison to treatment with IL-2/S4B6, which could contribute to the morbidity and mortality after LPS challenge to some extent (Figure 1 —figure supplement 2).”